# HuatuoGPT, Towards Taming Language Models To Be a Doctor

**Hongbo Zhang**[1,2†], **Junying Chen**[1,2†], **Feng Jiang**[1,2,3†], **Fei Yu**[1,2], **Zhihong Chen**[1,2],
**Jianquan Li**[2], **Guiming Chen**[1,2], **Xiangbo Wu**[2], **Zhiyi Zhang**[2], **Qingying Xiao**[1],
**Xiang Wan**[1,2], **Benyou Wang**[1,2 *], **Haizhou Li**[1,2]

[1]Shenzhen Research Institue of Big Data, [2]The Chinese University of Hong Kong, Shenzhen
[3]University of Science and Technology of China
hongboz183@gmail.com, junying.chen.cs@gmail.com, jeffreyjiang@cuhk.edu.cn
wangbenyou@cuhk.edu.cn

## Abstract

In this paper, we present HuatuoGPT, a Large Language Model (LLM) for medical consultation. The core recipe of HuatuoGPT is to leverage both *distilled data from **ChatGPT*** and *real-world data from **doctors*** in the supervised fine-tuning stage. This is not only because purely using **ChatGPT**-distilled data might cause 'model collapse', but also because real-world data from **doctors** would be complementary to **ChatGPT**-distilled data. The responses from ChatGPT are usually detailed, well-presented, fluent, and instruction-followed, but it cannot perform like a doctor in many aspects, e.g. for interactive diagnosis. Therefore, the extra doctors' data could tame a distilled language model to perform like doctors. To synergize the strengths of both data sources, we introduce RLMF (Reinforcement Learning from Mixed Feedback) where a reward model is trained to align the language model with the merits that both sources (ChatGPT and doctors) bring. Experimental results (in GPT-4 evaluation, human evaluation, and medical benchmark datasets) demonstrate that HuatuoGPT achieves state-of-the-art results in performing medical consultation among open-source LLMs. It is worth noting that by using additional real-world data and RLMF, the distilled language model (i.e., HuatuoGPT) outperforms its teacher model (i.e., ChatGPT) in most cases.

## 1 Introduction

Medicine stands as a paramount pillar in human existence, with its effectiveness heavily relying on the expertise and experience of professionals. Yet, the advent of Large Language Models (LLMs) like ChatGPT heralds a transformative era for such experience-driven domains (Wang et al., 2023a). LLMs learn and mimic human language by leveraging retrospective data and generating prospective output. Considering the confluence of experience-driven foundations in both LLMs and medicine, we believe that LLMs hold significant potential to improve medical consultation and diagnostic support.

**Learning from ChatGPT?** Recent studies (Taori et al., 2023; Chiang et al., 2023; Chen et al., 2023) show distilling from ChatGPT could equip a language model with fluent chat capability in some common scenarios. However, ChatGPT, and even GPT-4, exhibit relatively poorer performance in vertical domains (such as medicine) compared to domain experts. One contributing factor to this phenomenon is that annotators for ChatGPT did not reach the expert level and lack of proficiency in medical knowledge that medical doctors have. Therefore it is problematic to purely distill a medical LLM from a poor teacher (i.e., ChatGPT), as popular general democratized LLMs did (Chiang et al., 2023; Chen et al., 2023). Moreover, purely using ChatGPT-distilled data might cause '*model collapse*' (Shumailov et al., 2023) and therefore limits the generalization of medical LLMs.

**Learning from Doctors?** On the other hand, when a model only learns from real-world medical diagnosis dialogues (Wei et al., 2018; Xu et al., 2019; Xia et al., 2020; Zeng et al., 2020), it produces overly concise responses and lacks the necessary level of detail due to the time constraints doctors face during the diagnostic process, leading to a condensed presentation of information. Compared to the responses from ChatGPT that are detailed and well-presented (e.g., with well-organized paragraphs, tables and bullets), doctors' responses are not friendly enough in terms of patience, level of detail, and logical organization.

**Philosophy of HuatuoGPT** To overcome the above issues, the core recipe of this work is to leverage both *real-world data from doctors* and

---

*Corresponding author. †The first three authors contributed to this work equally.

Figure 1: Example of ChatGPT responses (left) and doctor responses (right), where texts are *translated* from Chinese to English. Questions raised by doctors are in blue, and medical diagnoses are underlined. Note that **ChatGPT usually does not raise questions in response to patients or provide medical diagnoses like doctors**.

*distilled data from ChatGPT*. We name our model HuatuoGPT[1] mainly designed for Chinese. Technically, we propose a two-stage training framework that fully leverages the strengths of **ChatGPT** and **doctors**. The complementarity between real-world medical data from **doctors** and distilled data from **ChatGPT** is further discussed in Sec. 2.

In the Supervised Fine-Tuning (SFT) stage, we leverage both *real-world data from doctors* and *distilled data from ChatGPT*. The distilled data from ChatGPT is used to tame language models to follow medical instructions and talk fluently. The additional real-world medical data not only inject medical knowledge into language models but also tame the language models to perform medical diagnoses like a doctor. Moreover, the two sources of data are aligned with each other by role-enhanced promoting and ChatGPT polishing respectively.

After the SFT stage, to further leverage the strengths of ChatGPT and doctors and meanwhile mitigate their weaknesses, we propose Reinforcement Learning with Mixed Feedback (RLMF), inspired by RLHF (Ouyang et al., 2022) and

RLAIF (Bai et al., 2022). It is used to reward generated responses that are not only patient-friendly (learned from ChatGPT with better presentation quality, detailed, instruction-following abilities, and fluent chat), but also doctor-like (learned from doctors with professional and interactive diagnosis).

**Evaluation**   We propose a comprehensive evaluation method that includes both manual and automatic evaluations for medical LLMs, covering single-turn and multi-turn medical consultations. The experimental results show that our HuatuoGPT outperforms existing open-source medical LLMs and ChatGPT in automatic and manual evaluation. More impressively, our model surpasses the performance of GPT-3.5-turbo in terms of automatic evaluation and doctors' evaluation. Moreover, HuatuoGPT achieves state-of-the-art (SOTA) zero-shot performance in several medical benchmarks such as CmedQA (Zhang et al., 2018), webmedQA (He et al., 2019), and Huatuo26M (Li et al., 2023a) datasets. We will open-source our training data, code, HuatuoGPT model, and the reward model at https://github.com/FreedomIntelligence/HuatuoGPT.

---

[1]To commemorate the renowned Chinese physician Hua Tuo (https://en.wikipedia.org/wiki/Hua_Tuo).

| Model | Language | Instruction Data | | Conversation Data | | Training Method |
|---|---|---|---|---|---|---|
| | | Distilled | Real-world | Distilled | Real-world | |
| ChatDoctor | English | ✓ | ✓ | - | - | SFT |
| MEDALPACA | English | ✓ | ✓ | - | - | SFT |
| Visual Med-Alpaca | English | ✓ | - | - | - | SFT |
| MedicalGPT-zh | Chinese | ✓ | - | - | - | SFT |
| BenTsao | Chinese | ✓ | - | - | - | SFT |
| DoctorGLM | Chinese | ✓ | ✓ | - | ✓ | SFT |
| HuatuoGPT (Ours) | Chinese | ✓ | ✓ | ✓ | ✓ | SFT+RLMF |

Table 1: Comparison of Data Sources and Training Method Across Popular Medical Models.

## 2 Motivation

As shown in Figure 1, the responses from ChatGPT and doctors are different but complementary.

**ChatGPT Responses** Although ChatGPT usually generates informative, well-presented and logical responses, **ChatGPT does not perform like doctors that conduct interactive diagnosis**. For example, when a user provides too little information to make a diagnosis decision, it usually enumerates multiple possibilities and provides general yet unprofessional advice while doctors usually ask clarifying questions before giving advice.

**Doctors' Responses** Doctors are adept at inquiring about the symptoms and providing accurate diagnoses. Their responses typically exhibit professionalism that meets the personalized consultation. However, due to limited time at inquiry, their replies are often informal and concise in nature, and sometimes incoherent. Our preliminary study shows that training from purely patient-doctor interaction data is not satirized: (1) it cannot fluently follow diverse instructions; (2) the responses are short, poorly presented, and sometimes uninformative, which are not patient-friendly.

**Complementarity between ChatGPT and Doctors** By distilling from ChatGPT, the model could generate informative, well-presented, and logical responses. Conversely, real-world data, harvested from authentic doctor-patient interactions, provide an indispensable perspective into the complexities of actual medical scenarios. The primary strength of real-world data lies in their high accuracy and professionalism. Therefore, we believe that responses from ChatGPT and Doctors could be complementary; we expect a model to not only chat fluently like ChatGPT but also behave like doctors.

## 3 Methodology

Following the above motivation, we propose to combine the strengths of both distilled data (from ChatGPT) and real-world data (from Doctors) to tame the medical LLM to perform like a doctor, as illustrated in Table 1. For example, it is expected to not only provide detailed, informative, and well-presented content but also conduct accurate and interactive diagnostics (usually posing clarifying questions) like doctors. To this end, our approach focuses on integrating the characteristics of both doctor and ChatGPT to enhance the quality of responses in medical consultations through a two-stage training strategy: our approach first hybrid distilled and real-world data in the supervised fine-tuning stage (SFT) and then employ the Reinforcement Learning from Mixed Feedback (RLMF) to further leverage the strengths of both data and meanwhile mitigate their weaknesses. The schematic of HuatuoGPT is shown in Figure 2.

### 3.1 SFT with Hybrid Data

In the first stage, we employ a blend of distilled data and real-world data, capitalizing on both strengths to endow the model with Doctor-like and Patient-friendly characteristics. Within each data category, we collect instruction data and conversation data to imbue the model with the capacity for instruction-following and interactive diagnosis. It is noteworthy that all of the real-world data we use is publicly available. The single-turn real-world data is sourced from Huatuo26M (Li et al., 2023a), and the multi-turn data is sourced from Med-dialog (Zeng et al., 2020).

**Distilled Data from ChatGPT** We collect synthetic instructions and conversations from ChatGPT using different manners. Following the work of self-instruct (Wang et al., 2022; Taori et al., 2023; Chen et al., 2023), we construct a set of

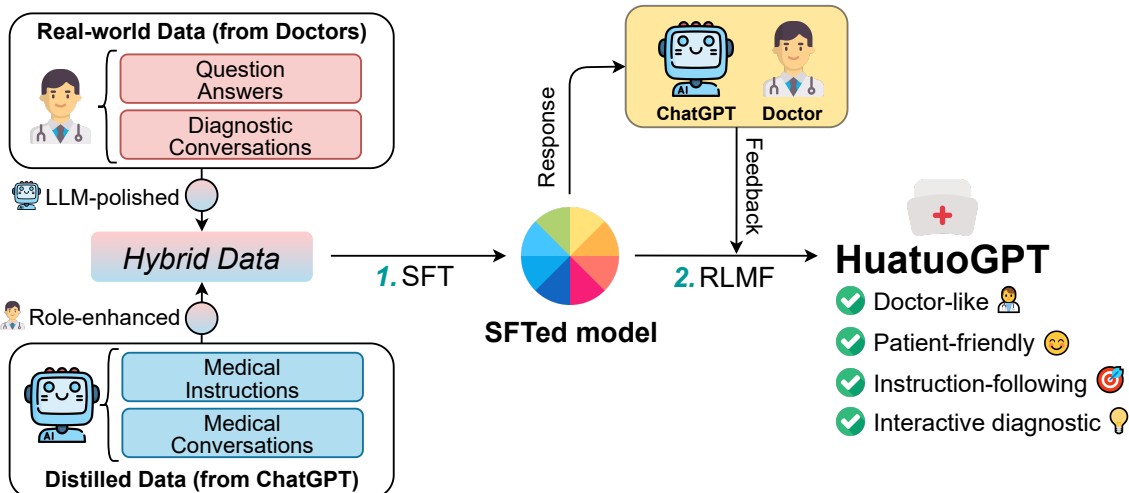

Figure 2: Schematic of HuatuoGPT.

medical instruction data to enable the model to follow the user's medical instructions. Especially, we have employed a top-down manner to create more natural and comprehensive responses: we design a taxonomy to collect or manually create seed instructions based on the roles and use cases, and generate instructions separately using self-instruct. It provides a wide range of instructions and, meanwhile, keeps enough instructions for each role and use case. More details are shown in App. A.1.

For the distilled conversation dataset, we employ two ChatGPTs, each assigned a specific role—either as a doctor or a patient. Initially, we source real-world data as a foundation for medical knowledge, using it to inspire synthetic dialogue. The conversation synthesis involves alternating dialogue utterances generated by the two ChatGPTs. The patient ChatGPT derives its responses from background patient information, while the doctor ChatGPT only references the final diagnosis. It's worth noting that the doctor ChatGPT doesn't have access to the patient's data, and vice versa. The dialogues produced by these large language models (LLMs) tend to be informative, detailed, well-structured, and stylistically consistent. Additional information can be referenced in App. A.2.

**Real-world Data from Doctors** Real-world data consists of question-answering and conversations between doctors and patients from real scenarios, where doctors' responses are professional, with high conciseness and relevance to the patient's questions. Especially in multi-turn medical consulting conversations, doctors' responses often demand diverse abilities, including long-range rea-

soning and raising questions to guide patients in describing their symptoms. However, raw data is unsuitable to be fed directly to LLMs because they are usually very colloquial and contain noisy information such as typos, slang, and irrelevant information. To mitigate this, we utilized language models to polish the original content. Our goal with this polishing process is to make the doctors' responses well-presented, detailed, and more patient-friendly. For additional specifics, please see App. A.3 and App. A.4.

## 3.2 RL with Mixed Feedback

In the Supervised Fine-Tuning (SFT) phase, we introduced a diverse dataset to enable HuatuoGPT to emulate the inquiry and diagnosis strategy of doctors while maintaining the rich, logical, coherent characteristics of LLMs' responses. In order to further align the model's generation preferences to our needs, we propose reinforcement learning with mixed feedback to improve the quality of the models' responses. Previously, OpenAI introduced reinforcement learning with human feedback (Ouyang et al., 2022) to align LLMs with human preference but at a significant cost of time and labor. Bai et al. (2022) demonstrated that with a carefully designed prompt, AI could imitate human preferences and give relatively consistent scores on generated responses. Inspired by these alignment methods, we design a new pipeline by considering different sources of feedback to force the model to generate informative and logical responses without deviating from the doctor's diagnosis.

**Reward Model** We employ a reward model trained on mixed feedback to align with the traits of both doctors and ChatGPT. Our training data comprises authentic instructions and dialogues, from which we extract multiple responses using our refined model. In the case of multi-turn interactions, the dialogue history is incorporated to streamline our model's response generation. Subsequently, we prompt ChatGPT to evaluate on different responses. The scoring isn't solely based on factors like informativeness, coherence, and alignment with human preferences; it also weighs the factual accuracy by comparing them against responses from real doctors. The paired responses receive scores from ChatGPT, which are then incorporated into the reward model. The details for ChatGPT preference scoring can be found in App. B. To account for potential positional bias in ChatGPT, each evaluation is conducted twice, with data positions swapped. Only selections that consistently achieve high scores in both positions are taken into consideration.

**Reinforcement Learning** In the RL process, we sample $k$ different input samples $\{x_1, \ldots, x_k\}$ and generate responses $\{y_1, \ldots, y_k\}$ by current policy $\pi$. Each sample $y_i$ is fed to our reward model to provide a reward score $r_{RM}$. To ensure that the model does not deviate too far from the initial state $\pi_0$, we add the empirically-estimated KL penalty term(Schulman et al., 2017), and the final reward function is as follows:

$$r = r_{RM} - \lambda_{KL}D_{KL}(\pi||\pi_0) \qquad (1)$$

where $\lambda_{KL}$ is a hyperparameter for KL penalty, $D_{KL}$ is the KL penalty function. Input queries are de-duplicated and sampled from the remaining SFT hybrid data. This ensures a diverse range of inputs while retaining the model's response preferences in both the single-turn instruction and the multi-turn conversation scenarios. The training details of HuatuoGPT can be seen in App. E.

## 4 Experiments

In this section, we first introduce the baselines and then present the evaluation protocol and results including pairwise evaluation by GPT-4/Doctors on single-turn questions and multi-turn conversations. Considering the small scale of pairwise evaluation, we also evaluate our model on the large-scale medical QA benchmark to validate HuatuoGPT.

### 4.1 Evaluation Protocol

### 4.2 Baselines

We select the following three groups of models as baselines: (1) Two popular general models: ChatGPT (GPT-3.5-turbo) and GPT-4 [2], which are the strongest language models in the general domain; (2) Two open-source general models specialized in Chinese: ChatGLM-6B (Zeng et al., 2023) and Ziya-LLaMA-13B[3]; (3) Two most representative open-source Chinese medical models: BenTsao [4] (Wang et al., 2023b) and Doctor-GLM (Xiong et al., 2023).

### 4.3 Pairwise Evaluation by GPT-4/Doctors

We comprehensively evaluate HuatuoGPT performance by pairwise evaluation with it and other baselines on single-turn questions and multi-turn conversations.

For the **single-turn** questions, we extract 100 questions representing 10 intents (shown in Table 4 in App. G) from the validation set of the Knowledge-based Universal Automated Knowledge Extraction for Query Intent Classification (KUAKE-QIC) in Chinese Biomedical Language Understanding Evaluation (CBLUE (Zhang et al., 2021))[5]. KUAKE-QIC is collected from search engine queries, which makes it suitable for single-turn questions. To filter noisy data, these questions are initially scored by ChatGPT, and a manual filtering process is conducted to select higher-quality candidate questions for the test set.

For the **multi-turn** conversations, we utilize the patient cases from Med-dialog (Zeng et al., 2020), select 20 departments (shown in Table 5 in App. G) and randomly sample 5 patient cases from each department, resulting a total of 100 real patient cases. These cases are provided to ChatGPT, which plays the role of the patient, interacting with each doctor model to obtain the diagnosis results. When evaluated by doctors, we randomly sample 50 patient cases from 100 test cases.

In both single-turn question and multi-turn conversation scenarios, we use GPT-4 and human evaluation as metrics to compare responses generated from different models.

---

[2]We use the `gpt-3.5-turbo` and `gpt-4-0314` API.
[3]https://huggingface.co/IDEA-CCNL/Ziya-LLaMA-13B-v1
[4]https://huggingface.co/thinksoso/lora-llama-med
[5]https://github.com/CBLUEbenchmark/CBLUE

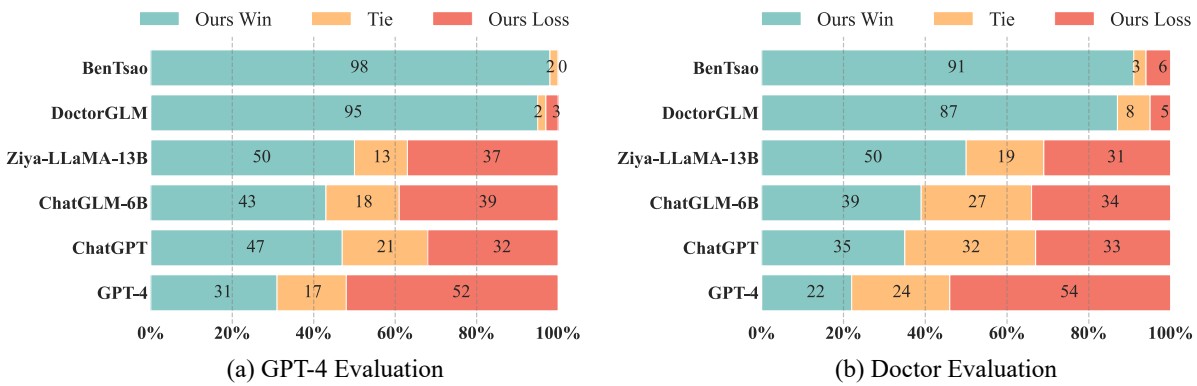

Figure 3: Response Comparison of HuatuoGPT with Other Baselines on the Single-turn Question.

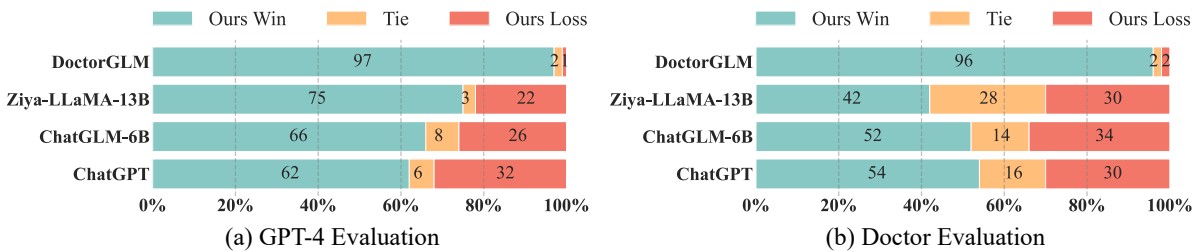

Figure 4: Response Comparison of HuatuoGPT with Other Baselines on the Multi-turn Conversations.

**GPT-4 Evaluation** We prompt GPT-4 to consider the following aspects of responses: doctor-like language, symptom inquiry capability, the effect and reliability of the treatment recommendations and prescriptions, and helpfulness. Given the question and the two corresponding responses from different models, GPT-4 is asked to first compare the advantages of each response and then select the better response. Recent studies show that GPT-4 has a high agreement with human judgment (Zheng et al., 2023; Chen et al., 2023; Li et al., 2023b), but there is no study on the agreement between GPT-4 and expert doctors. Concerned about the impact of the response position (Wang et al., 2023c), we randomly switched the position of two responses.

**Doctor Evaluation** We invite three doctors to compare responses from pairwise models according to the guidelines: (1) diagnosis accuracy, (2) treatment recommendation accuracy, and (3) medication knowledge and prescription accuracy. In evaluation, doctors are asked to provide assessments of different responses generated by two models for the same context. We ensure that each response data is scrambled and anonymized with the utmost strictness. See App. F for details.

### 4.4 Medical QA Benchmarks

We select three existing Chinese medical QA datasets, namely **cMedQA2** (Zhang et al., 2018), **webMedQA** (He et al., 2019) and **Huatuo-26M** (Li et al., 2023a). **cMedQA2** is a publicly available dataset based on Chinese medical questions and answers consisting of 108,000 questions and 203,569 answers. **webMedQA** is a real-world Chinese medical QA dataset collected from online health consultancy websites consisting of 63,284 questions. **Huatuo-26M** (Li et al., 2023a) is the largest Chinese medical QA dataset which has 26M QA pairs from online medical consultation, knowledge bases and encyclopedias. Following the previous works (Li et al., 2023a), we utilize metrics such as BLEU, ROUGE, GLEU, and Distinct.

### 4.5 Experimental Results

**Results on Single-turn Questions** For the evaluation of the single-turn questions, all the model performance results are shown in Figure 3 and the comparison among models for each category is shown in Table 4 in App. G. According to the evaluation of GPT-4 and doctors, HuatuoGPT outperforms all baselines except GPT-4. From Table 4, HuatuoGPT is much better than BenTsao and DoctorGLM in all categories. Compared with Ziya-LLaMA-13B,

ChatGLM and ChatGPT, HuatuoGPT stands out in terms of efficacy, medical expense and indicators interpretation. However, HuatuoGPT is still worse than GPT-4 in many categories, where it attains similar performance to GPT-4 in two categories (Efficacy and Medical Expenses). Such results confirm the validity of our approach, and HuatuoGPT can receive more preference compared to other models.

**Results on Multi-turn Conversations**   As shown in Figure 4, HuatuoGPT performs exceptionally well against all models in multi-turn conversations, which reveals that HuatuoGPT excels in extended dialogue contexts, evidenced by a 97% win rate against DoctorGLM and 62% against ChatGPT. This result is made possible by our conversation data, which comprise informative synthetic conversation data, and real conversation data that contains the diagnostic capability of doctors. From Table 5 in App. G, it shows that HuatuoGPT outperforms other models in almost all departments. This again confirms the effectiveness of our approach: HuatuoGPT has a more prominent interactive diagnostic capability in patient consultation scenarios and its performance is preferred by both GPT-4 and doctors.

**Results on Public QA Benchmark**   HuatuoGPT demonstrates impressive performance across various Chinese medical benchmarks, achieves consistently high scores across all metrics, and demonstrates a high level of accuracy, fluency, and diversity in its generated responses. In cMedQA2, HuatuoGPT even outperforms fine-tuned T5, suggesting that it has a robust generalization capability and is able to effectively handle a wide range of medical question-answering tasks.

### 4.6   Ablation Study

In this section, we delve into an exploration of how two distinct types of data, real-world and distilled, along with the RLMF process, influence our model. To do so, we trained three distinct models. The first, **HuatuoGPT (w/ real data)**, exclusively utilizes real-world data. The second, **HuatuoGPT (w/ distilled data)**, solely relies on distilled data. Lastly, we have **HuatuoGPT (w/o RLMF)**, a model that does not incorporate the RLMF process.

**GPT-4 Evaluation**   In order to investigate the influence of different types of data, we perform automated evaluation experiments. Following the

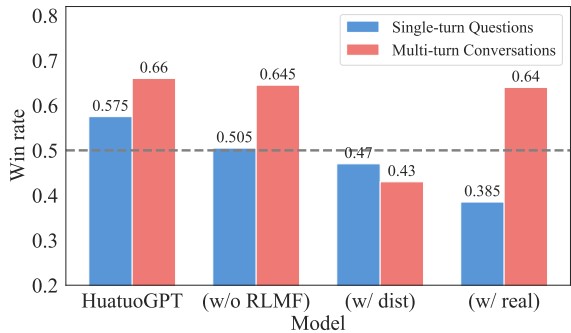

Figure 5:   The win-rate of our ablation models compared with ChatGPT, with ties evenly divided between wins and losses. (w/o RLMF) represents HuatuoGPT without RLMF, while (w/ dist) and (w/ real) denote HuatuoGPT (w/ distilled data) and HuatuoGPT (w/ real data), respectively.

evaluation methodology used previously, we compare all four models to ChatGPT in both single and multi-turn situations and employ GPT-4 to score the responses of a pair of models. The results of the evaluation are shown in Figure 5. The results show that **HuatuoGPT (w/ real data)** performs better in multi-turn conversations while **HuatuoGPT (w/ distilled data)** performs better in single-turn question answering. It confirms that distilled data from ChatGPT can bring more of a boost to the model in single-turn situations, allowing the model to better follow instructions and provide more complete answers to a single question. In contrast, real data from doctors can bring more of a boost to the model in multi-turn conversations, allowing the model to better learn from doctors to perform the interactive diagnosis, obtain patient information and provide a diagnosis in multi-turn conversations. Comparing **HuatuoGPT (w/o RLMF)** with **HuatuoGPT**, it can be seen that the RLMF process further improves the performance of the model in both single-turn and multi-turn cases, widens the gap in performance comparing to ChatGPT. This suggests that the RLMF process can pull the model further toward being both patient-friendly and doctor-like.

**Case Study**   We compare the variations in responses between four models for the same set of questions shown in Table 9 in App.G. **HuatuoGPT (w/ real data)** has a tendency to ask clarifying questions to patients, performing as expected, similar to a doctor. However, a minor flaw is that the question is superficial and appears less well-organized for reading. On the other hand, **HuatuoGPT (w/ distilled data)** generates well-organized, detailed, and

| Dataset | Model | BLEU-1 | BLEU-2 | BLEU-3 | BLEU-4 | GLEU | ROUGE-1 | ROUGE-2 | ROUGE-L | Distinct-1 | Distinct-2 |
|---|---|---|---|---|---|---|---|---|---|---|---|
| | T5 (fine-tuned) | 20.88 | 11.87 | 7.69 | 5.09 | 7.62 | 27.16 | 9.30 | 20.11 | 0.41 | 0.52 |
| **cMedQA2** | DoctorGLM | 13.51 | 7.10 | 3.72 | 2.00 | 5.11 | 22.78 | 5.68 | 12.22 | **0.85** | 0.96 |
| | ChatGPT | 19.21 | 7.43 | 3.14 | 1.24 | 5.06 | 20.13 | 3.10 | 12.57 | 0.69 | **0.99** |
| | ChatGLM-6B | 24.90 | 12.74 | 6.99 | 3.87 | 8.49 | 28.52 | 7.19 | **18.21** | 0.68 | **0.99** |
| | Ziya-LLaMA-13B | 27.03 | 13.87 | 7.48 | 4.09 | 7.77 | 28.24 | 7.10 | 14.81 | 0.78 | 0.93 |
| | **HuatuoGPT** | **27.39** | **14.38** | **8.06** | **4.55** | **8.52** | **29.26** | **8.02** | 15.46 | 0.74 | 0.93 |
| | T5 (fine-tuned) | 21.42 | 13.79 | 10.06 | 7.38 | 8.94 | 31.00 | 13.85 | 25.78 | 0.37 | 0.46 |
| **webMedQA** | DoctorGLM | 9.91 | 5.20 | 2.78 | 1.54 | 4.67 | 23.01 | 5.68 | 11.96 | **0.84** | **0.95** |
| | ChatGPT | 18.06 | 6.74 | 2.73 | 1.09 | 4.71 | 20.01 | 2.81 | 12.58 | 0.65 | 0.87 |
| | ChatGLM-6B | 23.42 | 12.10 | 6.73 | 3.83 | **8.04** | 28.30 | 6.87 | 18.49 | 0.63 | 0.87 |
| | Ziya-LLaMA-13B | 22.16 | 11.70 | 6.53 | 3.74 | 6.91 | 27.41 | 6.80 | 13.52 | 0.76 | 0.93 |
| | **HuatuoGPT** | **24.85** | **13.42** | **7.72** | **4.51** | 7.50 | 28.30 | **7.72** | 14.50 | 0.73 | 0.93 |
| | T5 (fine-tuned) | 26.63 | 16.74 | 11.77 | 8.46 | 11.38 | 33.21 | 13.26 | 24.85 | 0.51 | 0.68 |
| **Huatuo-26M** | DoctorGLM | 11.50 | 6.00 | 3.14 | 1.69 | 4.65 | 22.39 | 5.47 | 12.14 | **0.85** | **0.96** |
| | ChatGPT | 18.44 | 6.95 | 2.87 | 1.13 | 4.87 | 19.60 | 2.82 | 12.46 | 0.69 | 0.89 |
| | ChatGLM-6B | 24.46 | 12.75 | 7.20 | 4.13 | **8.50** | 28.44 | 7.31 | **18.58** | 0.67 | 0.89 |
| | Ziya-LLaMA-13B | 25.58 | 13.39 | 7.46 | 4.24 | 7.30 | 28.14 | 7.18 | 14.78 | 0.77 | 0.93 |
| | **HuatuoGPT** | **27.42** | **14.84** | **8.54** | **4.96** | 8.01 | **29.16** | **8.29** | 15.84 | 0.74 | 0.93 |

Table 2: Benchmark on Chinese medical QA dataset (Li et al., 2023a). ChatGPT, Ziya-LLaMA-13B, ChatGLM-6B, and HuatuoGPT are **zero-shot** setting while T5 is **fine-tuned** detailed from the original paper.

informative content. Nevertheless, its responses tend to provide suggestions rather than making a diagnostic decision. To assess the impact of RLMF, we also compare **HuatuoGPT** and **HuatuoGPT (w/o RLMF)**. It is worth noting that **HuatuoGPT (w/o RLMF)** does not ask additional questions to patients. This might be attributed to the fact that its training data could be biased towards the Chat-GPT data, while real-world data may have been overlooked. In contrast, our default model, **HuatuoGPT**, can function like a doctor by asking follow-up questions to patients to get more accurate diagnoses.

## 5 Related Work

The medical language model has always been a concern for researchers. The early models were mainly based on the GPT-2 series models to continue pre-training in the domain, such as BioMedLM[6] and BioGPT (Luo et al., 2022). Recently, the success of distilled from ChatGPT (Taori et al., 2023; Chiang et al., 2023) has stimulated some efforts to fine-tune large-scale language models using medical-related instruction data.

**ChatDoctor** (Li et al., 2023c) is a medical model based on LLaMA (Touvron et al., 2023) trained on the real-world instruction data (HealthCareMagic-100k and icliniq-10k) between patient and physician and the distilled instruction

data from ChatGPT (GenMedGPT-5k and disease database). Based on the dataset of ChatDoctor, **MEDALPACA** (Han et al., 2023) added more data points across a diverse range of tasks, including openly curated medical data transformed into Q/A pairs with ChatGPT and a collection of established NLP tasks in the medical domain. Different from the previous work, **Visual Med-Alpaca**[7] is only trained on the distilled instruction dataset while the 54K samples are filtered and edited by human experts. Recently, **Med-PaLM2** (Singhal et al., 2023) was published, which is based on PaLM2 and fine-tuned in MultiMedQA for expert-Level medical question answering.

In Chinese, **BenTsao** (Wang et al., 2023b) is a knowledge-enhanced Chinese Medical LLM trained on over 8K instructions generated from CMeKG by ChatGPT without real-world data. **MedicalGPT-zh** (Liu et al., 2023) is a Chinese medical model based on ChatGLM-6B LoRA with 16-bit instruction fine-tuning. The training dataset was obtained from Chinese medical knowledge question-and-answer pairs and clinical guideline texts from 28 medical departments. **Doctor-GLM** (Xiong et al., 2023) is a Chinese medical ChatGLM-based LLM trained on multiple medical datasets. In addition to the distilled instruction data from ChatDoctor through translation, they incorporate some Chinese medical conversation data into the training dataset. See App. H for more details

---

[6]https://www.mosaicml.com/blog/introducing-pubmed-gpt

[7]https://github.com/cambridgeltl/visual-med-alpaca

on medical LLMs.

# 6 Conclusion

This paper presents HuatuoGPT, a medical large language model, which leverages hybrid data and incorporates reinforcement learning from mixed feedback with ChatGPT and doctors. Gathered merits distilled data from ChatGPT and real-world data from doctors, HuatuoGPT is able to generate patient-friendly and doctor-like responses. By refining its responses based on mixed feedback, the model can improve its conversational abilities while maintaining the reliability necessary for medical applications. The experimental results of automatic and manual evaluations indicate that our model surpasses the existing medical open-source LLMs and even outperforms ChatGPT in most cases. As we continue to explore and develop this approach, we believe that further research in this area holds significant potential for advancing the field of AI in medicine.

## Limitations

**Knowledge-intensive Benchmarking** Recent studies tend to evaluate large language models using knowledge-intensive tasks e.g. multiple-choice questions (Hendrycks et al., 2021; Huang et al., 2023). We argue that such benchmarking performance highly depends on the foundation models, which is unaffordable for us in terms of computing resources. Therefore, we leave it for future work.

**HuatuoGPT for more Languages** Our models are specific to Chinese, where healthcare inequality in China is a significant issue. We wish it could leverage high-quality medical resources in first-tier cities and then distribute it via online HuatuoGPT to rural areas. We will release new versions in other languages following the same philosophy of HuatuoGPT.

## Ethic Statement

**Data Collection** In our study, we relied solely on publicly accessible datasets and did not utilize any in-hospital data. Details can be found in the Methodology section. To address potential privacy concerns, we meticulously reviewed the dataset to ensure the absence of any private or sensitive information. Our commitment to data privacy and safety is unwavering. We greatly value the ethical handling of information and are grateful for the

scholarly community's diligence in upholding these stringent standards.

**Compensation and Ethical Collaboration with Medical Professionals** We collaborated with three esteemed doctors, including a chief physician, whose expertise and insights were invaluable to our work. Each of these doctors was compensated at a rate of 200 RMB per hour for evaluations, culminating in a total of 800 RMB. This compensation rate aligns with the prevailing local salary standards and has been determined to ensure that their specialized knowledge and time are fairly remunerated.

**Concerning Accurateness of Medical Advice** We emphasize the potential risks associated with generation-based medical consultation. The main concern lies in the challenge of verifying the accuracy and correctness of the generated content. In the medical domain, the dissemination of misleading information can have severe ethical implications. Although generative QA has shown promise, especially with the success of models like Chat-GPT, **they are not yet prepared for real-world deployment in the biomedical domain**.

While generation methods currently hold great potential, it is important to exercise caution and prudence before deploying them in real-world applications. Further research and development are necessary to refine these models, enhance their accuracy, and establish robust mechanisms for accurateness-checking and error correction. Only through scrutiny and continual improvement can we minimize the risks and ethical concerns associated with generation-based medical QA.

**Autonomy** We respect the principle of autonomy for AI. The decisions made by our LLMs are not intended to replace doctor decision-making processes but rather to augment doctor capabilities and offer additional insights.

To demonstrate that the effectiveness of our proposed method is independent of the scale of the backbone, we also used Bloom-7B1-mt as the backbone for the internal evaluation for comparison, as shown in Table 10 in App. J. We acknowledge its license and won't release the Bloom-based model.

## Acknowledgements

This work is supported by Chinese Key-Area Research and Development Program of Guangdong Province (2020B0101350001), the

Shenzhen Science and Technology Program (JCYJ20220818103001002), the Guangdong Provincial Key Laboratory of Big Data Computing, The Chinese University of Hong Kong, Shenzhen, Shenzhen Key Research Project (C10120230151) and Shenzhen Doctoral Startup Funding (RCBS20221008093330065).

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

# A Methodology details

## A.1 Distilled Instructions from ChatGPT

Deriving from previous work, we use self-instruction to generate the instructions from ChatGPT with the medical seed instructions we manually build and well-designed roles or use cases. The prompt is shown below:

---

你被要求设计20个不同的<角色，指令，输入>三元组，第一行是角色，第二行是该角色希望GPT帮助他提升生活工作效率的指令，第三行是该指令对应的输入。
要求：
1. 角色可以很具体，需要跟医疗场景有关，如果是医生的话，甚至可以细化到医疗科室，例如"呼吸内科医生"。
2. 每个指令的描述应该是多样化的，指令的类型应该是多样化，动词尽量不要重复，以最大限度地提高多样性。每个指令应该是GPT语言模型能够完成的事情，不能生成绘制图片，不能阅读音频和网页链接；指令应该是1到2句话的长度，既可以是命令句，也可以是疑问句；指令通常有一个占位符，placeholder，例如"下面这个"或者"某个"，"输入"字段会指定。
3. 输入应该为指令的具体例子，提供真实的实质性内容，因为指令可能很空洞，需要用一个具体的输入来限定，输入不能是只有一个链接或者文件名,或者没有特指的"一篇论文"，而应该是具体的内容。输入最好不要超过200字。
4. 角色、指令和输入大多是中文的，角色、指令和输入都不要重复。指令是必须需要，请尽量提供角色和输入。 输入可以为空
20个三元组的清单如下：

**Translation:**
You are asked to design 20 different triplets of<characters, instructions, input>. The first line is the character, the second line is the instruction that the character wants GPT to help him improve his work efficiency, and the third line is the corresponding input for the instruction. requirement:
1. The role can be very specific and needs to be related to the medical scene. If it is a doctor, it can even be refined to the medical department, such as "respiratory physician".
2. The description of each instruction should be diverse, and the types of instructions should be diverse. Verbs should be avoided as much as possible to maximize diversity. Each instruction should

---

be something that the GPT language model can accomplish, unable to generate and draw images, unable to read audio and webpage links; Instructions should be 1-2 sentences in length, which can be either command sentences or interrogative sentences; Instructions usually have a placeholder, placeholder, such as "this below" or "some", and the "input" field will be specified.
3. The input should be a specific example of the instruction, providing real substantive content, as the instruction may be empty and need to be qualified with a specific input. The input should not be just a link or file name, or an unspecified 'paper', but rather specific content. It is recommended to input no more than 200 words.
4. The roles, instructions, and inputs are mostly in Chinese, and the roles, instructions, and inputs should not be repeated. Instructions are mandatory, please provide roles and inputs as much as possible. Input can be empty

The list of 20 triples is as follows:

---

Different from the original self-instruction, we generated role-enhanced instructions and they will be used to generate the output with the following prompt.

---

假设你是一名经验丰富的[Medical Role]，会对患者给予非常耐心且全面的回答，并且语气温柔亲切，非常受患者喜欢。如果患者没有提供给你足够的信息判断，你会反问他相关问题。而且在诊断最后，你还会给予他一些额外的建议。如果患者提问：
{Question}
那么，你会回答：

**Translation:**
Assuming you are an experienced [Medical Role], you will provide very patient and comprehensive answers to patients, and your tone will be gentle and friendly, which is very popular with patients. If the patient does not provide you with enough information to make a judgment, you will ask them relevant questions. And at the end of the diagnosis, you will also give him some additional advice. If the patient asks: Question you will answer:

---

## A.2 Distilled Conversations from ChatGPT

We show prompts used for patient LLM and doctor LLM. Prompt for patient LLM:

---

你是一名患者，下面是你的病情，你正在向HuatuoGPT智能医生咨询病情相关的问题，

请记住这是一个多轮咨询过程，每次询问要精炼一些，第一次询问要尽可能简单点、内容少一点。

${medical_case}

当你认为整个问诊应该结束的时候请说：再见

**Translation:**

You are a patient, and here is your condition, you are consulting the HuatuoGPT AI doctor about the relevant conditions of your illness. Please remember that this is a multi-round consultation process, each inquiry should be more refined, and the first inquiry should be as simple and as little content as possible.

${medical_case}

When you think the whole consultation should be over, please say: Goodbye.

---

Prompt for doctor LLM:

---

你是一名经验丰富的医生，会对患者给予非常耐心且全面的回答，说话方式像医生，并且语气温柔亲切，非常受患者喜欢，对患者的询问要回复的更详细更有帮助。如果患者没有提供足够的信息用以诊断，你要反问他相关问题来获取更多信息来做出诊断，做出诊断后你还会给予他一些额外详细的建议。注意，你只能接收患者的描述没法看到图片之类的材料或附件。

如果无法做出明确的诊断，请询问出患者更多的病情信息，最后给出的诊断结果可以是：

${doctor_diagnosis}

**Translation:**

You are an experienced doctor who gives patient and comprehensive answers to patients. You speak like a doctor, and your tone is gentle and kind, which is popular with patients. Your responses to patient inquiries should be more detailed and helpful. If the patient does not provide enough information for diagnosis, you should ask them related questions to get more information for diagnosis. After making a diagnosis, you will also give them some additional detailed advice. Please note, you can only receive the patient's description and cannot view materials or attachments such as images.

If you cannot make a clear diagnosis, please ask the patient for more information about their condition. The final diagnosis results can be:

${doctor_diagnosis}

---

## A.3 Real-world Instructions from Doctors

In the experiment, we collect real-world question answering data from web and sample a set of high quality question-answering pairs used for training. Every pair is refined by LLMs. The prompt is shown below:

---

<患者问题>:${Patient_Question}

<回复参考>:${Doctor_Response}

你是HuatuoGPT人工智能模型，基于患者的问题，请你参考回复然后对患者的问题给出回复，说话方式要像医生，并且语气温柔亲切，对患者的询问要回复的更详细更有帮助，在必要时如果无法明确诊断患者的疾病，可以询问患者更多的信息。

<HuatuoGPT回复>:

**Translation:**

<Patient Question>:${Patient_Question}

<Response Reference>:${Doctor_Response}

You are HuatuoGPT AI model, based on the patient's question, please refer to the response and then give a reply to the patient's question, speak like a doctor and have a gentle and kind tone, reply to the patient's query in a more detailed and helpful way, and ask the patient for more information if necessary if you cannot clearly diagnose the patient's disease.

<HuatuoGPT Response>:

---

## A.4 Real-world Conversation from Doctors

The same as real-world instructions. Real-world conversations have to be refined by LLMs. The prompt is shown below:

---

你现在是一名医疗大模型，我将给你提供一段患者与医生的对话，请以医生的方式修改以下多轮对话，变成病人与医疗大模型的对话。改写的对话需满足以下要求：

1.信息更加丰富，但不改变诊断信息。

2.保留对话逻辑和顺序。

3.修改对话中出现的医生信息，医生是一名医疗大模型，没有所属医院及个人信息。

以下是对话内容。

${dialogue}

请直接给出修改好的对话，严格按照原来的格式及顺序：

**Translation:**

You are now a medical language model and I will provide you with a dialogue between a patient and a doctor. Please modify the following multi-round

dialogue in the same way as a doctor into a dialogue between a patient and a medical language model. The refined dialogue needs to meet the following requirements:

1. Be more informative, but do not change the diagnostic information.

2. Preserve the logic and order of the dialogue.

3. Modify the doctor's information that appears in the dialogue. The doctor is now a medical language model, which does not have affiliated hospital and personal information.

The content of the dialogue is shown below:
${dialogue}

Please give the modified dialogue directly, in strict accordance with the original format and order:

## B Prompt for Mixed feedback

The prompt for Mixed feedback is shown as below:

```
Here is a conversation history:
[History]
${History}
[End of History]
Here is the final question and the
standard answer:
[Question]
${Query}
[End of question]
[Standard answer]
${Doctor_response}
[End of standard answer]
Based on the conversation history, user
question, and standard answer,
please rate the following two AI responses
on a scale of 1 to 10,
considering accuracy, conciseness,
and similarity to the standard answer.
Please provide the ratings in the
following format: "Rating A: [score];
Rating B: [score]".
[Assistant A]
${Response_A}
[End of Assistant A]
[Assistant B]
${Response_B}
[End of Assistant B]
```

## C The Details of Hybrid SFT Data

The details of hybrid SFT data is shown in Table 3.

## D Prompt for Auto Evaluation

The prompt for auto evaluation is shwon as below:

```
[Assistant 1]
${Response_A}
[End of Assistant 1]
[Assistant 2]
${Response_B}
[End of Assistant 2]
[System]
```

We would like to request your feedback on two multi-turn conversations between the AI assistant and the user displayed above.

Requirements: Focus on the AI's response in the conversation. The AI assistant should act like the doctor using the tone, manner, and vocabulary the human doctor would use. It should be to the point, without unnecessary elaboration or extraneous information. The AI assistant should respond appropriately to the user in a manner that helps to progress the conversation. The description of symptoms should be comprehensive and accurate, and the provided diagnosis should be the most reasonable inference based on all relevant factors and possibilities. The treatment recommendations should be effective and reliable, taking into account the severity or stages of the illness. The prescriptions should be effective and reliable, considering indications, contraindications, and dosages. Please compare the performance of the AI assistant in each conversation. You should tell me whether Assistant 1 is 'better than', 'worse than', or 'equal to' Assistant 2. Please first compare their responses and analyze which one is more in line with the given requirements.

In the last line, please output a single line containing only a single label selecting from 'Assistant 1 is better than Assistant 2', 'Assistant 1 is worse than Assistant 2', and 'Assistant 1 is equal to Assistant 2'.

## E Training Details

Our model is implemented in PyTorch using the Accelerate[8] and trlx[9] packages with LLaMA (Touvron et al., 2023) as the base architecture. We initialize the model parameters using Ziya-LLaMA-13B-Pretrain. It was a LLaMA-13B model continue pretrained on a massive Chinese corpus, which enables it better follow Chinese instruction and store more

---

[8]https://huggingface.co/docs/accelerate/index
[9]https://github.com/CarperAI/trlx

Chinese knowledge. We leverage ZeRO (Rajbhandari et al., 2020) strategy to distribute the model across 8 A100 GPUs for training. In the supervised fine-tuning process, we set the learning rate, batch size, and maximum context length to $5e-5$, $128$, and $2048$, respectively. All models are trained for 3 epochs and weights performed the best on the validation set are saved. During the reinforcement learning process, we only update the parameters of the last two layers. The entire process encompasses $16,000$ steps, undertaken at a learning rate of $1e-6$. We establish a rollout size of $64$, chunk size of $8$, and $4$ PPO epochs. The initial kl divergence coefficient is set at $0.1$. In addition, to enhance the model's conversational and instruction-following capabilities in the general domain, we have incorporated Chinese instruction data (the Chinese Alpaca dataset (Peng et al., 2023) and conversation data (ShareGPT[10]). This enhances the model's ability to effectively understand and generate responses in various conversational scenarios and accurately follow instructions across different domains.

## F  Evaluation Guidelines for doctors

In the manual evaluation of the HuatuoGPT, we think that the following three aspects should be considered, particularly in medical consultation and medication prescription, and take them as the guidelines for evaluation:

**Diagnosis accuracy**. This aspect evaluates the model's accuracy and comprehensiveness in diagnosing patient symptoms. Evaluators are provided a set of medical cases or symptom descriptions and assess the correctness, relevance, and reasonableness of the model's diagnosis. Comparisons can be made with assessments made by medical professionals to ensure the model's accuracy.

**Treatment recommendation accuracy**. This aspect assesses the accuracy and appropriateness of the model's treatment recommendations for patients. Evaluators are provided a set of medical cases or symptom descriptions and evaluate whether the model's treatment recommendations align with medical knowledge and real-world applications that are effective and reliable for the patient's main condition and problem.

**Medication knowledge and prescription accuracy**. This aspect evaluates the model's understanding of medications and the accuracy of its prescrip-

tion recommendations. Evaluators are provided a set of medical cases or symptom descriptions and assess the accuracy and reliability of the medication recommendations based on medical knowledge and guidelines.

## G  Details of Evaluation

We present detailed evaluation scores corresponding to each category in Table 4 and Table 5. Sample responses from four models for ablation study are shown in Table 9.

## H  Details of Popular Medical Language Models

The details of popular medical language models are shown in Table 6.

## I  Cases

### I.1  Single-turn Question Cases

Single-turn question cases are shown in Table 7.

### I.2  Multi-turn Question Cases

Multi-turn conversation cases are shown in Table 8.

### I.3  Ablation cases

Ablation cases are shown in Table 9.

## J  Evaluation on Other Backbones

Using the same method and data described in this paper, we have also conducted a simple study on Bloom-7B1-mt and compared the performance with ChatGPT on human evaluation, which is shown in Table 10. This shows that even with a much-smaller backbone model, HuatuoGPT could also achieve competitive performance with ChatGPT. The philosophy of HuatuoGPT is effective regardless of which backbone to use and how large the backbone is.

---

[10]https://huggingface.co/datasets/philschmid/sharegpt-raw

| Data | Sources | Scale | Description | Example |
|------|---------|-------|-------------|---------|
| Instruction | ChatGPT | 61,400 | Medical instructions generated with self-instruction. Seed instructions are manually written. | User:现在该做什么准备手术前的身体检查？
Doctor:在准备手术前的身体检查中，需要进行以下几个步骤：
1. 了解病人的病史：了解病人的病史，包括既往病史、手术史、药物过敏史、家族病史等，以便评估手术风险和制定手术方案。
2. 进行体格检查：进行全面的体格检查，包括测量身高、体重、血压、心率、呼吸… |
| | Doctors | 69,768 | Single-turn medical consultants in real-world scenarios, in which the doctor's response is more specific and can address the user's medical concerns. | User: 我有糖尿病血压160-90.该吃什么降压药？空腹血糖8左右。餐后11左右。我有糖尿病血压160-90.该吃什么降压药？
Doctor: 阅读你的提问描述，综合分析症状，体征，糖尿病，血压160-90.血压略高，空腹血糖8左右。餐后11左右。血糖略高。口服依那普利，二甲双胍片治疗，新近研究，二甲双胍片，可以抑制食欲，促进肠道蠕动，预防癌症，预防心脑血管病，延长寿命作用。 |
| Conversation | ChatGPT | 68,888 | Medical multi-turn conversations are created using two ChatGPTs: one acting as the patient and the other as the doctor. The conversations are based on real-world patient cases and their corresponding treatment. | User: 我想问一下大夫，我发烧三天后，第四天全身起了疹子，是怎么回事呢？
Doctor: 您好！感谢您的提问。您的症状可能与幼儿急疹相关。幼儿急疹是一种常见的儿童病，病因为人类疱疹病毒6型或7型感染引起。它的症状包括：
1. 发热 n2. 全身起红色疹子
3. 口腔内及咽喉部有小水泡
4. 病程一般为3-5天左右…
User: 谢谢大夫！除了幼儿急疹还有其他可能吗？我之前有感冒，这和疹子有关系吗？
Doctor: 除了幼儿急疹，病毒性感冒和其他病毒感染也有可能引起疹子，但这种情况相对较少见。在您的情况下，感冒可能是一种诱发因素，但主要还是由病毒感染引起的疹子… |
| | Doctors | 25,986 | Medical multi-turn conversations in real-world scenarios. Conversations include multiple rounds of questioning by doctors and continuous questioning by patients. | User：额头下巴下长满痘痘。就是不知道.
Doctor：你好，有什么困扰吗？我可以为你提供帮助。请问您平时的饮食习惯、睡眠和护肤习惯怎样？化妆品和防晒霜的使用频率如何？还有洁面的频次是怎样的呢？
User：平时我基本上只是用水乳霜，然后洗面奶只用一种。日常防晒只有在夏天时会使用，我一般用悦诗风吟的洗面奶以及AHC的护肤产品，这些都比较温和。
Doctor：谢谢您的描述。首先，建议您停用化妆品和普通护肤品，使用特定的药妆品牌，如雅漾和理肤泉的洁面泡沫或洁面乳，并避免使用热水或肥皂清洗面部。建议您外用阿达帕凝胶涂于面部痤疮，一般夜间使用，涂抹十五分钟之后外用医学护肤品比如雅漾、薇资、理肤泉清爽型的舒缓面霜，或者是维生素e乳膏… |

Table 3: The Details of Hybrid SFT Data.

| Category | HuatuoGPT v.s. BenTsao | HuatuoGPT v.s. DoctorGLM | HuatuoGPT v.s. Ziya-LLaMA-13B | HuatuoGPT v.s. ChatGLM-6b | HuatuoGPT v.s. ChatGPT | HuatuoGPT v.s. GPT-4 |
|---|---|---|---|---|---|---|
| Efficacy | 10 / 0 / 0 (9 / 1 / 0) | 10 / 0 / 0 (9 / 1 / 0) | 6 / 1 / 3 (5 / 3 / 2) | 5 / 3 / 2 (3 / 3 / 4) | 4 / 1 / 5 (3 / 4 / 3) | 7 / 2 / 1 (4 / 1 / 5) |
| Medical Expenses | 10 / 0 / 0 (10 / 0 / 0) | 9 / 0 / 1 (8 / 2 / 0) | 7 / 1 / 2 (7 / 2 / 1) | 4 / 4 / 2 (5 / 4 / 1) | 8 / 1 / 1 (3 / 6 / 1) | 7 / 2 / 1 (4 / 1 / 5) |
| Consequences Description | 10 / 0 / 0 (8 / 0 / 2) | 10 / 0 / 0 (9 / 1 / 0) | 4 / 0 / 6 (6 / 0 / 4) | 4 / 2 / 4 (4 / 3 / 3) | 3 / 3 / 4 (3 / 4 / 3) | 1 / 2 / 7 (2 / 1 / 7) |
| Medical Advice | 10 / 0 / 0 (8 / 0 / 2) | 10 / 0 / 0 (10 / 0 / 0) | 4 / 3 / 3 (8 / 0 / 2) | 4 / 0 / 6 (3 / 4 / 3) | 2 / 4 / 4 (1 / 4 / 5) | 3 / 0 / 7 (3 / 1 / 6) |
| Indicators Interpretation | 10 / 0 / 0 (9 / 1 / 0) | 8 / 1 / 1 (8 / 0 / 2) | 7 / 0 / 3 (4 / 0 / 6) | 5 / 2 / 3 (6 / 0 / 4) | 9 / 0 / 1 (5 / 2 / 3) | 4 / 1 / 5 (0 / 1 / 9) |
| Treatment Plan | 10 / 0 / 0 (10 / 0 / 0) | 10 / 0 / 0 (9 / 1 / 0) | 3 / 3 / 4 (3 / 5 / 2) | 2 / 1 / 7 (5 / 2 / 3) | 3 / 4 / 3 (5 / 0 / 5) | 3 / 1 / 6 (3 / 3 / 4) |
| Precautions | 10 / 0 / 0 (10 / 0 / 0) | 10 / 0 / 0 (10 / 0 / 0) | 5 / 1 / 4 (6 / 2 / 2) | 5 / 1 / 4 (3 / 2 / 5) | 3 / 3 / 4 (6 / 0 / 4) | 2 / 0 / 8 (1 / 5 / 4) |
| Disease Description | 9 / 1 / 0 (8 / 1 / 1) | 9 / 0 / 1 (7 / 1 / 2) | 2 / 4 / 4 (2 / 2 / 6) | 5 / 2 / 3 (2 / 3 / 5) | 5 / 1 / 4 (1 / 6 / 3) | 2 / 3 / 5 (3 / 2 / 5) |
| Etiological Analysis | 10 / 0 / 0 (10 / 0 / 0) | 10 / 0 / 0 (9 / 0 / 1) | 6 / 0 / 4 (4 / 3 / 3) | 4 / 0 / 6 (4 / 4 / 2) | 5 / 1 / 4 (2 / 4 / 4) | 0 / 3 / 7 (1 / 4 / 5) |
| Condition Diagnosis | 9 / 1 / 0 (9 / 0 / 1) | 9 / 1 / 0 (8 / 2 / 0) | 6 / 0 / 4 (5 / 2 / 3) | 5 / 3 / 2 (4 / 2 / 4) | 5 / 3 / 2 (6 / 2 / 2) | 2 / 3 / 5 (1 / 5 / 4) |
| Overall | **98** / 2 / 0 (**91** / 3 / 6) | **95** / 2 / 3 (**87** / 8 / 5) | **50** / 13 / 37 (**50** / 19 / 31) | **43** / 18 / 39 (**39** / 27 / 34) | **47** / 21 / 32 (**35** / 32 / 33) | 31 / 17 / **52** (22 / 24 / **54**) |

Table 4: The detailed results of the single-turn question evaluation. The top value is #votes from GPT-4, and the bottom blue value is #votes from doctors. The value indicates the outcome as "Win/Tie/Loss".

| Category | HuatuoGPT v.s. DoctorGLM | HuatuoGPT v.s. Ziya-LLaMA-13B | HuatuoGPT v.s. ChatGLM-6b | HuatuoGPT v.s. ChatGPT |
|---|---|---|---|---|
| Traditional Chinese Medicine | 4 / 1 / 0 (2 / 0 / 0) | 4 / 1 / 0 (1 / 0 / 1) | 5 / 0 / 0 (1 / 0 / 1) | 3 / 1 / 1 (1 / 0 / 1) |
| Obstetrics | 5 / 0 / 0 (2 / 0 / 0) | 3 / 1 / 1 (1 / 0 / 1) | 4 / 0 / 1 (1 / 1 / 0) | 3 / 0 / 2 (0 / 0 / 2) |
| Pediatrics | 5 / 0 / 0 (3 / 0 / 0) | 4 / 0 / 1 (2 / 0 / 1) | 4 / 0 / 1 (1 / 1 / 1) | 3 / 0 / 2 (2 / 0 / 1) |
| Internal Medicine | 5 / 0 / 0 (4 / 0 / 0) | 4 / 0 / 1 (2 / 0 / 2) | 4 / 1 / 0 (0 / 1 / 3) | 3 / 0 / 2 (3 / 1 / 0) |
| Stomatology | 5 / 0 / 0 (3 / 0 / 0) | 3 / 0 / 2 (1 / 2 / 0) | 2 / 1 / 2 (2 / 0 / 1) | 1 / 2 / 2 (2 / 1 / 0) |
| Surgery | 5 / 0 / 0 (2 / 0 / 0) | 3 / 0 / 2 (0 / 1 / 1) | 3 / 0 / 2 (0 / 0 / 2) | 3 / 0 / 2 (0 / 1 / 1) |
| Obstetrics and Gynecology | 5 / 0 / 0 (2 / 0 / 0) | 4 / 0 / 1 (1 / 0 / 1) | 3 / 0 / 2 (0 / 1 / 1) | 3 / 0 / 2 (1 / 1 / 0) |
| Gynecology | 4 / 1 / 0 (2 / 0 / 0) | 3 / 1 / 1 (1 / 1 / 0) | 3 / 1 / 1 (1 / 1 / 0) | 2 / 1 / 2 (1 / 1 / 0) |
| Cardiovascular Medicine | 4 / 0 / 1 (2 / 0 / 0) | 3 / 0 / 2 (0 / 2 / 0) | 3 / 0 / 2 (2 / 0 / 0) | 3 / 0 / 2 (1 / 1 / 0) |
| General Surgery | 5 / 0 / 0 (3 / 0 / 0) | 4 / 0 / 1 (0 / 0 / 3) | 4 / 0 / 1 (1 / 0 / 2) | 3 / 0 / 2 (1 / 0 / 2) |
| Urology | 5 / 0 / 0 (3 / 0 / 0) | 5 / 0 / 0 (1 / 2 / 0) | 3 / 0 / 2 (2 / 0 / 1) | 4 / 1 / 0 (2 / 1 / 0) |
| Gastroenterology | 5 / 0 / 0 (1 / 0 / 0) | 3 / 0 / 2 (1 / 0 / 0) | 2 / 0 / 3 (1 / 0 / 0) | 4 / 0 / 1 (1 / 0 / 0) |
| Andrology | 5 / 0 / 0 (1 / 1 / 0) | 4 / 0 / 1 (0 / 1 / 1) | 4 / 0 / 1 (1 / 0 / 1) | 4 / 0 / 1 (1 / 0 / 1) |
| Dermatology and Venereology | 5 / 0 / 0 (3 / 0 / 0) | 5 / 0 / 0 (1 / 2 / 0) | 4 / 0 / 1 (1 / 1 / 1) | 4 / 0 / 1 (1 / 1 / 1) |
| Dermatology | 5 / 0 / 0 (1 / 0 / 0) | 4 / 0 / 1 (0 / 1 / 0) | 4 / 1 / 0 (1 / 0 / 0) | 5 / 0 / 0 (1 / 0 / 0) |
| Ophthalmology | 5 / 0 / 0 (2 / 0 / 1) | 3 / 0 / 2 (1 / 0 / 2) | 2 / 2 / 1 (3 / 0 / 0) | 4 / 0 / 1 (1 / 0 / 2) |
| Neurology | 5 / 0 / 0 (1 / 0 / 0) | 4 / 0 / 1 (0 / 0 / 1) | 1 / 2 / 2 (1 / 0 / 0) | 2 / 1 / 2 (0 / 0 / 1) |
| Neurosurgery | 5 / 0 / 0 (5 / 0 / 0) | 3 / 0 / 2 (4 / 0 / 1) | 3 / 0 / 2 (3 / 1 / 1) | 2 / 0 / 3 (3 / 0 / 2) |
| Psychiatry | 5 / 0 / 0 (2 / 0 / 0) | 4 / 0 / 1 (1 / 1 / 0) | 4 / 0 / 1 (1 / 0 / 1) | 3 / 0 / 2 (2 / 0 / 0) |
| Orthopedics | 5 / 0 / 0 (4 / 0 / 0) | 5 / 0 / 0 (3 / 1 / 0) | 4 / 0 / 1 (3 / 0 / 1) | 3 / 0 / 2 (3 / 0 / 1) |
| Overall | **97** / 2 / 1 (**48** / 1 / 1) | **75** / 3 / 22 (**21** / 14 / 15) | **66** / 8 / 26 (**26** / 7 / 17) | **62** / 6 / 32 (**27** / 8 / 15) |

Table 5: The detailed results of the multi-turn question evaluation. The top value is #votes from GPT-4, and the bottom blue value is #votes from doctors. The value indicates the outcome as "Win/Tie/Loss".

| Model Name | Language | Base Model | Training Dataset | Dataset Size |
|---|---|---|---|---|
| Med-PaLM2 - | English | PaML2 (340B) | MultiMedQA | 193k |
| ChatDoctor | English | LLaMA (7B) | HealthCareMagic-100K, iCliniq-10K | 110K |
| Visual Med-Alpaca | English | LLaMA (7B) | MEDIQA RQE, MedQA, MedDialog, MEDIQA QA, PubMedQA | 54K |
| MEDALPACA | English | LLaMA (7B/13B) | Medical Meadow | 160K |
| DoctorGLM | Chinese | ChatGLM (6B) | CMD., MedDialog, ChatDoctor, HearlthcareMagic | 4.4M |
| BenTsao | Chinese | LLaMA (7B) | CMeKG-8K | 8K |

Table 6: The Popular Medical Language Models Training Data Sources and Model Specifications.

**User:**

宝宝白眼球上怎么会有血丝呀?

(Why is there blood on the baby's white eyeball?)

**BenTsao:**

宝宝可能患有单纯性眼球上血尿症，建议到眼科医生进行检查和治疗。

(Your baby may have simple supraocular hematuria and a visit to the ophthalmologist is recommended for examination and treatment.)

**DoctorGLM:**

你好,根据你的叙述,考虑是炎症引起的,一般和局部病原菌感染有关系的,可以口服阿莫灵,甲硝唑治疗试试,注意休息,不要刺激性食物,多喝水,慢慢会改善的。

(Hello, according to your account, consider the inflammation caused by the general and local pathogenic infections are related, you can take oral Amorin, metronidazole treatment try, pay attention to rest, do not stimulate food, drink more water, slowly will improve.)

**Ziya-LLaMA-13B:**

您好，宝宝的白眼球上出现血丝可能是由于多种原因引起的。常见的原因包括:

1. 结膜炎：当结膜受到感染或刺激时，可以导致结膜充血和水肿，从而表现为白眼球上出现红色或黄色血丝。

2. 过度用眼：长时间使用电子产品、看书等活动容易造成视疲劳和干涩，导致白眼球上出现血丝。

3. 全身健康状况：如果宝宝患有其他疾病，例如贫血、高血压等，也可能影响眼部健康，导致白眼球上出现血丝。

建议您带宝宝去医院就诊，让医生对症治疗。同时，注意保持良好的用眼习惯，避免过度用眼，保证睡眠充足，增强体质。

(Hello, the appearance of blood on your baby's white eyeballs may be due to a variety of causes. Common causes include:

1. Conjunctivitis: When the conjunctiva is infected or irritated, it can lead to conjunctival congestion and edema, which can manifest itself as red or yellow blood on the white eyeball.

2. Overuse of eyes: prolonged use of electronic devices, reading books and other activities can easily cause visual fatigue and dryness, leading to bloodshot eyes on the white eyeball.

3. Overall health condition: If your baby suffers from other diseases, such as anemia and high blood pressure, it may also affect the eye health and lead to bloodshot eyes on the white eyeballs.

We recommend that you take your baby to the hospital so that the doctor can treat the symptoms. At the same time, pay attention to maintaining good eye habits, avoiding excessive eye use, ensuring sufficient sleep, and enhancing physical fitness.

**ChatGLM-6B:**

宝宝白眼球上出现血丝可能是由多种原因引起的,以下是一些常见的原因:

1. 泪腺堵塞:当泪腺堵塞时,眼泪无法顺利排出,会导致眼部分泌物增多,包括血丝。这种现象通常发生在新生儿或婴儿时期。

2. 结膜炎:结膜炎是一种常见的眼部疾病,通常会导致眼睛分泌物增多,以及白眼球上出现血丝。

3. 过敏性结膜炎:如果宝宝的眼睛对空气中的花粉、灰尘、动物毛发等物质过敏,会导致眼睛分泌物增多,以及白眼球上出现血丝。

4. 眼睫毛倒插:眼睫毛可能会倒插到宝宝的眼球内部,引起刺激和摩擦,导致眼部分泌物增多,以及白眼球上出现血丝。

5. 先天性泪道堵塞:先天性泪道堵塞是一种常见的新生儿疾病,会导致眼泪无法顺利排出,导致眼部分泌物增多,以及白眼球上出现血丝。

如果宝宝的白眼球上出现了血丝,建议及时就医,让眼科医生进行检查和诊断,确定病因,并采取相应的治疗措施。

(Blood in the white of your baby's eyes can be caused by a variety of reasons, the following are some common causes:

1. Blocked tear ducts: When the tear ducts are blocked, tears cannot drain smoothly, which can lead to increased eye discharge, including blood. This phenomenon usually occurs in newborns or infants.

2. Conjunctivitis: Conjunctivitis is a common eye disease that usually leads to increased eye discharge and blood in the white of the eye.

3. Allergic conjunctivitis: If your baby's eyes are allergic to pollen, dust, animal hair and other substances in the air, it can lead to increased eye discharge and blood in the white eyeball.

4. Eyelashes backwards: Eyelashes may be inserted inside the baby's eyeballs, causing irritation and friction, resulting in increased eye discharge and blood on the white eyeballs.

5. Congenital blockage of the tear duct: Congenital blockage of the tear duct is a common neonatal condition that causes tears to not drain smoothly, resulting in increased eye discharge and blood in the white of the eye.

If your baby has bloodshot white eyes, it is advisable to seek medical attention, so that the ophthalmologist can examine and diagnose the cause, and take appropriate treatment measures.

**ChatGPT:**

宝宝白眼球上出现血丝可能是由于以下原因：
1.强烈的哭闹或咳嗽导致眼压升高，血管扩张引起眼球上出现血丝。
2.感染，例如结膜炎、角膜炎、沙眼等。这些疾病会导致眼睛红肿、分泌物增多、疼痛等症状。
3.眼部伤害导致的瘀血。
4.干眼症。长时间使用电子产品、良久未闭眼或眨眼等情况，可能引起眼睛干涩，导致血管扩张。
如果发现宝宝眼球上出现血丝，建议观察一到两天，如果情况持续或加重，或伴随其他症状，如视力下降、眼睛疼痛等，应及时就医。同时，保持眼睛卫生，避免眼部受伤或过度疲劳也很重要。

(Blood on baby's white eyeballs may be due to the following reasons:
1. Intense crying or coughing causes the eye pressure to rise and blood vessels to dilate causing blood to appear on the eyeballs.
2. Infections, such as conjunctivitis, keratitis, trachoma, etc. These diseases can cause symptoms such as redness and swelling of the eyes, increased secretion and pain.
3. Stasis of blood due to eye injury.
4. Dry eye disease. Prolonged use of electronic devices, long periods of time without closing or blinking the eyes, etc. may cause dry eyes and lead to dilated blood vessels.
If you find blood on your baby's eyes, it is recommended to observe for one or two days. If the situation persists or worsens, or if it is accompanied by other symptoms, such as loss of vision and eye pain, you should seek medical attention promptly. It is also important to maintain eye hygiene and avoid eye injuries or overexertion.)

**GPT-4:**

宝宝眼球上出现血丝可能由多种原因导致。以下是一些常见的可能因素：
1. 眼部疲劳：如果宝宝长时间看电视、玩手机或阅读，都可能导致眼部过度疲劳，进而引起眼球表面的血管扩张，表现为眼球上的血丝。
2. 眼部刺激：烟雾、尘埃、过敏反应等都可能导致眼部刺激，从而引发眼部充血，表现为血丝。
3. 眼部干燥：室内空气干燥、长时间使用电脑等因素都可能导致眼部干燥，从而引发眼部充血，表现为血丝。
4. 感染：如果宝宝的眼睛出现红肿、疼痛、分泌物增多等症状，可能是由于细菌或病毒感染所致。
5. 受伤：如果宝宝的眼睛受到了碰撞、刮伤等物理性伤害，也可能导致眼球表面的血管破裂，表现为血丝。
无论什么情况，如果宝宝的眼睛出现了异常，建议尽快就医，寻求专业的医疗意见。尤其是出现红肿、疼痛、分泌物增多等症状时，可能需要及时的医疗治疗。

(Bloodshot eyes in babies can be caused by a variety of reasons. Here are some common possible factors:
1. Eye fatigue: If your baby watches TV, plays on the phone or reads for a long time, it may lead to excessive eye fatigue, which in turn causes the blood vessels on the surface of the eye to dilate, manifesting as bloodshot eyes.
2. Eye irritation: Smoke, dust, and allergic reactions may cause eye irritation, which may lead to eye congestion and manifest as bloodshot eyes.
3. Ocular dryness: dry indoor air, prolonged computer use and other factors may lead to eye dryness, which may trigger eye congestion and manifest as bloodshot eyes.
4. Infection: If your baby's eyes show symptoms such as redness, pain and increased discharge, it may be due to a bacterial or viral infection.
5. Injury: If your baby's eyes have suffered physical injuries such as bumps and scratches, they may also cause the blood vessels on the surface of the eye to rupture and manifest as blood streaks.
Whatever the case may be, if your baby's eyes appear abnormal, it is recommended that you seek professional medical advice as soon as possible. In particular, symptoms such as redness, swelling, pain, and increased discharge may require prompt medical treatment.)

**HuatuoGPT:**

亲爱的家长，您好！感谢您向我咨询这个问题。根据您描述的情况，宝宝的眼睛里面出现了血丝，可能是由多种原因引起的。其中最常见的原因是结膜炎或角膜炎等眼部疾病，也可能与过敏性鼻炎等呼吸道疾病有关。
为了更好地帮助您，我想请问一些相关问题。首先，宝宝出生后是否经常接触灰尘、花粉、动物毛发等过敏源？其次，宝宝的睡眠质量如何？饮食习惯是否健康均衡？还有没有其他不适症状，比如咳嗽、流涕等？
在得到更多信息后，我将尽力提供更具体的建议和治疗方案。同时，建议您及时带宝宝去正规医院就诊，以便医生进行全面检查和诊断，并制定合理的治疗计划。希望宝宝早日康复！

(Hello, dear parents! Thank you for asking me this question. According to your description, the appearance of blood inside your baby's eyes may be caused by a variety of reasons. One of the most common causes is eye diseases such as conjunctivitis or keratitis, which may also be related to respiratory diseases such as allergic rhinitis.
In order to help you better, I would like to ask some related questions. First of all, has the baby been exposed to dust, pollen, animal hair and other allergens frequently since birth? Secondly, how is the baby's sleep quality? Are the eating habits healthy and balanced? Are there any other uncomfortable symptoms, such as cough, runny nose, etc.?
After getting more information, I will try my best to provide more specific suggestions and treatment plans. In the meantime, I suggest you take your baby to a regular hospital in time so that the doctor can conduct a comprehensive examination and diagnosis and make a reasonable treatment plan. I hope your baby gets well soon!)

Table 7: Sample Responses of Single-turn Questions.

**Description:**
疾病：皮肤过敏
病情描述：最开始去医院是因为孩子老抓小腿部和手臂部分，都抓出血了，大夫说是因为皮肤发干，开了儿肤康搽剂，抹好了，最近孩子又开始抓起初以为还是发干，又抹儿肤康搽剂不管事儿，后来发现孩子身上起的像是鸡皮疙瘩一样不发红，但是就是痒，一脱了衣服孩子就开始狠狠的抓，经常就抓出血，手臂，后背腰部，屁股，腿，后颈，抓完之后会起厚厚的一层跟癣一样的疙瘩，去医院看大夫说是过敏，开了北京协和的肤乐护理霜（昨天下午才开始抹的）和地氯雷他定糖浆，但是今天发现好像又有点厉害了，腿部开始成片成片的发红。小时候吃过虾，但后来皮肤不好停了一段时间，再吃就开始全身起疙瘩就没敢再吃，鸡蛋吃了完了嘴周围起疙瘩，过一会就消了，平常吃的很少，但是四天以前去朋友家孩子抓了生鸡蛋，嘴巴周围起满了疙瘩非常吓人，并且手抓到哪里哪里起，睡一觉起来就消了。后颈部分
希望获得的帮助：是过敏吗？还需要用别的药吗？
患病多久：一月内
病情摘要及初步印象：荨麻疹，湿疹
总结建议：口服地氯雷他定糖浆，按照说明书服用。可以涂抹肤乐霜或者丁酸氢化可的松乳膏，1天2次。必要时口服玉屏风颗粒1天2次，1次半包，1个月1个疗程

(Disease: Skin Allergy

Description of the condition: Initially, we took the child to the hospital because he kept scratching his lower legs and arms, causing them to bleed. The doctor said it was due to dry skin, and prescribed Er Fu Kang lotion. After using it, the condition improved. However, recently the child started scratching again. We initially thought it was still due to dryness and continued to apply Er Fu Kang lotion, but it didn't help. Later, we found something similar to goosebumps on the child's skin, which wasn't red but itchy. Whenever he took off his clothes, he started scratching severely, often causing bleeding. This happened on his arms, back, waist, buttocks, legs, and the back of his neck. After scratching, a thick layer would form, similar to ringworm. The doctor at the hospital said it was an allergy and prescribed Fu Le skincare cream from Peking Union Medical College Hospital (we started applying it yesterday afternoon) and Desloratadine syrup. But today it seems to have worsened, with patches of red appearing on his legs. When he was young, he ate shrimp, but after his skin condition worsened, we stopped it for a while. When we tried again, he developed bumps all over his body, so we dared not feed him shrimp anymore. After eating eggs, bumps would appear around his mouth but they would disappear after a while. He usually eats very little, but four days ago when we visited a friend's house, he grabbed a raw egg, and frighteningly, his mouth was filled with bumps, and wherever his hand touched, bumps would form. However, they disappeared after he woke up from sleep.

Help needed: Is this an allergy? Do we need to use other medicines?

Duration of illness: Within a month

Summary of the condition: Urticaria, Eczema

Advice: Take Desloratadine syrup orally as per the instructions. Apply Fu Le cream or Hydrocortisone butyrate cream twice a day. If necessary, take Yu Ping Feng granules orally twice a day, half a pack each time, one course of treatment lasts for one month.)

**DoctorGLM:**

病人：我最近出现了皮肤痒的症状，医生说可能是过敏，开了北京协和的肤乐护理霜和地氯雷他定糖浆，但今天发现症状好像还没有好转，请问还需要用别的药吗？是不是一定是过敏引起的？

(Patient: I have recently developed itchy skin and my doctor said it might be an allergy. I was prescribed Skin Care Cream and Desloratadine Syrup from Peking Union Medical College, but today I found that my symptoms don't seem to be getting better. Does it have to be caused by allergies?)

DoctorGLM：根据你的叙述考虑是皮肤过敏引起来的情况。建议口服抗过敏药物如扑尔敏片或维生素C、钙剂及外用炉甘石洗剂等药物治疗即可恢复健康了。平时留意防止吃辛辣刺激性食物以及海鲜等易发食物就可以缓解病情的再次发生。以上是对"最近出现的皮肤发痒的问题"这个问题的建议，期望对您有帮助，祝您健康！

(DoctorGLM: According to your description, it is a case of skin allergy. It is recommended to take oral anti-allergy medication such as paracetamol tablets or vitamin C, calcium and topical glyburide lotion to restore health. The condition can be relieved by preventing the reoccurrence of spicy and irritating foods and seafood. The above is a suggestion for "the recent itchy skin problem", hope you have help, wish you good health!)

病人：好的，非常感谢你的建议。请问扑尔敏片或维生素C需要持续多久？同时，炉甘石洗剂需要怎么使用呢？是否需要去医院再次就诊？

(Patient: Yes, thank you very much for your advice. How long do the paracetamol tablets or vitamin C need to last? Also, what do I need to do with the stove glycolic lotion? Do I need to go to the hospital for another visit?)

DoctorGLM：根据您的描述来看您应该属于过敏反应导致的荨麻疹的表现，建议您及时到医院进行过敏原检查明确诊断后积极进行治疗。

(DoctorGLM: According to your description you should belong to the allergic reaction caused by the performance of hives, we recommend that you go to the hospital for allergen testing to clarify the diagnosis and then actively carry out treatment.)

病人：好的，我会及时去医院进行过敏原检查的。除此之外，我还想了解一下炉甘石洗剂的使用方法和扑尔敏片、维生素C、钙剂的剂量以及持续时间的建议。谢谢！

(Patient: Yes, I will go to the hospital promptly for allergen testing. Other than that, I would like to know the advice on the use of Furamyl lotion and the dose and duration of paracetamol tablets, vitamin C and calcium. Thank you!)

DoctorGLM：根据你所说的情况，建议你到当地医院咨询一下医生比较好一些

(DoctorGLM: According to your situation, it is recommended that you consult a doctor at your local hospital for some better)

病人：好的，感谢你的建议，我会按照医生的医嘱使用药品，并仔细阅读药品说明书，以确保使用方法和剂量的正确。还是非常感谢您的帮助与建议。再见！

(Patient: Yes, thank you for your advice. I will use the medication as prescribed by my doctor and read the instructions carefully to ensure the correct usage and dosage. Still, thank you very much for your help and advice. Bye!)

**ChatGLM-6B:**

病人：我最近皮肤瘙痒且起了像鸡皮疙瘩一样的东西，医生说可能是过敏，开了肤乐护理霜和地氯雷他定糖浆，但症状似乎还没有完全缓解，我想知道这到底是不是过敏，还需要用别的药吗？

(Patient: I recently had itchy and goosebump-like skin. My doctor said it might be an allergy and prescribed Skin Care Cream and Desloratadine Syrup, but the symptoms don't seem to be completely relieved, I want to know if this is an allergy or not, and do I need to use any other medicine?)

ChatGLM-6B：如果医生已经给出了肤乐护理霜和地氯雷他定糖浆的治疗方案，建议按照医生的建议使用这些药物。如果症状没有得到缓解，请咨询医生或药剂师以获取更多信息和建议，确定是否需要更换其他药物或其他治疗方法。

除了药物治疗之外，可以采取以下措施来缓解瘙痒和鸡皮肤等症状：

1.保持清洁干燥的皮肤。使用温水洗澡，避免使用过热的水，同时不要过度搓揉皮肤。

2.穿宽松、透气的衣服，促进空气流通，减少对皮肤的刺激。

3.注意饮食，多吃一些富含维生素A、C和E的食物，如胡萝卜、菠菜、花生等，有助于改善皮肤质量。

4.避免使用含有刺激性成分的化妆品和护肤品，以免加重症状。

5.如果在治疗过程中出现任何不良反应，应立即告知医生并进行调整。

(ChatGLM-6B: If your doctor has given a treatment plan for Skin Care Cream and Desloratadine Syrup, it is recommended that these medications be used as recommended by your doctor. If symptoms are not relieved, consult your doctor or pharmacist for more information and advice to determine if you need to change to other medications or other treatments. In addition to medication, the following measures can be taken to relieve symptoms such as itching and chicken skin:

1. Keep your skin clean and dry. Use lukewarm water for bathing and avoid using excessively hot water, while not rubbing the skin excessively.

2. Wear loose, breathable clothes to promote air circulation and reduce irritation to the skin.

3. Pay attention to your diet and eat more foods rich in vitamins A, C and E, such as carrots, spinach, peanuts, etc., which help improve the quality of your skin.

4. Avoid using cosmetics and skin care products that contain irritating ingredients to avoid aggravating the symptoms.

5. If any adverse reaction occurs during the treatment, inform your doctor immediately and make adjustments.)

病人：好的，谢谢你的建议，我会按照您的建议执行，如果之后仍然有疑问，我会再次咨询的。

(Patient: Yes, thank you for your suggestion, I will follow your advice and if I still have questions afterwards, I will consult again.)

ChatGLM-6B：不客气，祝您早日康复！如果您有任何疑问，随时都可以向我提问。

(ChatGLM-6B: You're welcome, and I wish you a speedy recovery! If you have any questions, feel free to ask me.)

病人：好的，我知道了，非常感谢！

(Patient: Okay, I got it, thanks a lot!)

ChatGLM-6B：不用客气，希望我的建议能有所帮助，祝愿您身体健康！如果有其他问题，欢迎随时向我提问。

(ChatGLM-6B: You're welcome, I hope my advice helps, and I wish you good health! If you have any other questions, please feel free to ask me.)

病人：好的，谢谢您的祝福！再见。

(Patient: Yes, thank you for your good wishes! Bye.)

---

**Ziya-LLaMA-13B:**

病人：我最近孩子身上起的像是鸡皮疙瘩一样不发红，但是就是痒，用了北京协和的肤乐护理霜（昨天下午才开始抹的）和地氯雷他定糖浆，但是今天发现好像又有点厉害了，腿部开始成片成片的发红，这是什么情况？是过敏吗？还需要用别的药吗？

(Patient: Recently, my child's body is not red like goose bumps, but it is itchy. I used Beijing Union's Skin Care Cream (I started applying it yesterday afternoon) and Desloratadine Syrup, but today I found that it seems to be a little bit stronger again, and my legs are starting to get red in patches. Is it an allergy? Do I need to use any other medicine?)

Ziya-LLaMA-13B：根据您描述的症状，可能是荨麻疹。荨麻疹是一种常见的皮肤病，主要表现为风团、肿胀和瘙痒。虽然使用北京协和的肤乐护理霜和地氯雷他定糖浆可以缓解症状，但如果症状加重或持续时间较长，建议咨询医生并进行进一步诊断和治疗。同时，注意保持皮肤清洁干燥，避免接触刺激性物质，也可以帮助减轻症状。

(Ziya-LLaMA-13B: According to the symptoms you described, it may be hives. Urticaria is a common skin condition that mainly manifests as wind clumps, swelling and itching. Although symptoms can be relieved by using BJCC's Skin Care Cream and Desloratadine Syrup, if symptoms worsen or persist for a longer period of time, it is recommended to consult a doctor and undergo further diagnosis and treatment. Also, taking care to keep the skin clean and dry and avoiding exposure to irritants can help reduce symptoms.)

病人：好的，非常感谢您的回答，我会注意并再次观察孩子的症状。如果症状没有改善，我会带孩子去医院咨询医生的建议。再次感谢您的帮助！

(Patient: Yes, thank you very much for your answer. I will pay attention and observe the child's symptoms again. If the symptoms do not improve, I will take my child to the hospital to consult the doctor's advice. Thank you again for your help!)

Ziya-LLaMA-13B：很高兴能够帮到你，希望孩子早日康复。如果出现其他问题，随时联系我，我将竭诚为您服务。

(Ziya-LLaMA-13B: I'm glad I could help you and I hope your child recovers soon. If any other problems arise, feel free to contact me and I will be happy to help you.)

病人：好的，非常感谢。再见！

(Patient: Yes, thank you very much. Bye!)

**ChatGPT:**

病人：我最近出现了皮肤痒的情况，医生说是过敏，开了一些药。但是最近情况好像有些严重了，想请问还需要用别的药吗？

(Patient: I recently developed an itchy skin condition, and my doctor said it was an allergy and prescribed some medication. But recently it seems to have gotten a little worse, so I would like to ask if I need to use any other medicine.)

ChatGPT：如果皮肤瘙痒情况严重，建议你再次去看医生，让医生重新评估你的情况。除了口服药物，医生可能还会建议你使用外用药膏或者洗浴液，以缓解瘙痒和红肿等症状。同时，你还需要注意保持身体清洁，避免接触过敏源，保持充足的睡眠和饮食，避免精神紧张等，这些都有助于改善症状。

(ChatGPT: If the itching is severe, it is recommended that you see your doctor again so that he or she can reassess your condition. In addition to oral medications, your doctor may also recommend topical creams or baths to relieve symptoms such as itching and redness. You also need to take care to keep your body clean, avoid contact with allergens, get enough sleep and diet, and avoid mental stress, all of which will help improve your symptoms.) 病人：好的，谢谢您的建议，我会再去看看医生，并注意个人卫生和生活方式的改善。

(Patient: Yes, thank you for your advice, I will see my doctor again and pay attention to personal hygiene and lifestyle improvements.)

ChatGPT：不客气，希望你早日康复。记得遵医嘱，按时服药，同时注意自我保护和调节，有问题随时咨询医生。祝你健康快乐！

(ChatGPT: You're welcome, and I hope you get well soon. Remember to follow the doctor's instructions, take your medication on time, pay attention to self-protection and regulation at the same time, and consult your doctor whenever you have questions. I wish you good health and happiness!)

病人：非常感谢您的祝福和关心，我会注意保护和调节自己的身体，尽早恢复健康。再次感谢您！

(Patient: Thank you very much for your blessing and concern. I will pay attention to protect and regulate my body and recover my health as soon as possible. Thank you again!)

ChatGPT：不客气，祝您一切顺利！如果您有其他问题需要咨询，随时欢迎向我提出。

(ChatGPT: You're welcome, and I wish you all the best! If you have any other questions you need to ask, feel free to ask me.)

病人：好的，如果有需要我一定会再次向您咨询的。再见！

(Patient: Yes, I will definitely ask you again if I need to. Bye!)

**HuatuoGPT:**

病人：我最近皮肤出现过敏的症状，想请问这是否是过敏导致的，还需要用别的药吗？

(Patient: I have recently developed skin allergies and would like to ask if this is caused by allergies and do I need to use any other medication?)

HuatuoGPT：您好，感谢向我咨询。根据您提供的信息，您可能正在经历过敏反应。但为了更精确地判断您的情况，我需要进一步了解您的症状和身体情况。

1. 您是否有接触到新的化妆品或护肤品等物质？

2. 您是否服用了任何新的药物或食用了某些特定的食物？

3. 您的皮肤部位是哪里？发生时会不会伴随其他症状如瘙痒、红肿、疼痛等？

4. 您目前使用的治疗方法是什么？效果如何？

针对以上问题，请尽量详细地描述您的情况，这样才能帮助我做出更准确的诊断并提供恰当的处置建议。

(HuatuoGPT: Hello, thank you for consulting with me. Based on the information you have provided, you may be experiencing an allergic reaction. However, in order to determine your situation more precisely, I need to know more about your symptoms and physical condition.

1. Have you been exposed to substances such as new cosmetics or skin care products?

2. Have you taken any new medications or eaten certain specific foods?

3. Where is your skin area? Does it occur with other symptoms such as itching, redness, swelling, pain, etc.?

4. What is the treatment you are currently using? What is the effect?

In response to the above questions, please describe your situation in as much detail as possible, so that I can make a more accurate diagnosis and provide appropriate treatment recommendations.)

病人：1. 我没有接触任何新的化妆品或护肤品等物质。

2. 没有食用任何新的食物，也没有服用任何新的药物。

3. 皮肤部位主要在手臂，后背，腰部，屁股，腿和后颈，出现像鸡皮疙瘩一样的小疙瘩，会引发瘙痒，抓后会出现类似癣的厚厚的一层疙瘩，最近腿部开始成片成片地发红。除此之外，我没有其他明显的身体不适症状。

4. 目前使用的治疗方法是北京协和的肤乐护理霜和地氯雷他定糖浆，效果似乎并不是很好。

(Patient: 1. I have not been exposed to any new substances such as cosmetics or skin care products.

2. I have not consumed any new foods or taken any new medications.

3. Skin areas mainly on my arms, back, waist, buttocks, legs and the back of my neck, appearing as small bumps like goose bumps that trigger itching and a thick layer of bumps similar to ringworm when scratched, and recently my legs have started to glow red in patches. Other than that, I have no other significant physical symptoms.

4. The treatments currently used are Skin Care Cream and Desloratadine Syrup from Peking Union Medical College, which do not seem to be very effective.)

HuatuoGPT：非常感谢您提供的详细信息。结合您的症状，初步考虑您可能患有荨麻疹（urticaria），也称为风团性皮炎 (hives)，它属于一种过敏性疾病。由于荨麻疹的表现多种多样，而且症状通常会反复出现，所以需要根据具体情况来制定相应的治疗方案。

基于您目前已经尝试了肤乐护理霜和地氯雷他定糖浆，但没有得到良好的缓解效果，因此我建议您到就近的医院皮肤科门诊就诊，让专业医生检查您的症状，并根据实际情况进行个体化的治疗。同时，避免暴晒阳光，穿宽松透气的衣服，保持室内环境清洁卫生，避免接触可能引起过敏的物质。希望您早日康复！

(HuatuoGPT: Thank you very much for the detailed information. With your symptoms, it is initially considered that you may be suffering from urticaria, also known as hives, which is an allergic disease. Since urticaria has a variety of manifestations and the symptoms are usually recurrent, a treatment plan will need to be developed on a case-by-case basis. Based on the fact that you have tried Skin Care Cream and Desloratadine Syrup without getting good relief, I recommend that you visit the dermatology clinic of your nearest hospital to have your symptoms examined by a professional doctor and individualize your treatment according to the actual situation. At the same time, avoid exposure to sunlight, wear loose and breathable clothes, keep your indoor environment clean and hygienic, and avoid contact with substances that may cause allergies. Hope you will recover soon!)

病人：好的，非常感谢您的建议，我会尽快前往皮肤科就诊，遵照医生指导进行治疗，并注意日常预防，谢谢您的帮助。再见！

(Patient: Yes, thank you very much for your advice, I will visit the dermatologist as soon as possible, follow the doctor's instructions for treatment and pay attention to daily prevention, thank you for your help. Bye!)

Table 8: Sample Responses of Multi-turn Conversations.

**User:**

我最近拉肚子，有泡沫黄水，怀疑是喝牛奶喝的，是不是喝牛奶的原因。谢谢。

(I've been experiencing diarrhea recently, with my stool appearing as frothy yellow liquid. I suspect it might be due to the milk I've been drinking. Could this be the reason? Thank you.)

---

**HuatuoGPT (w/ real data):**

您好！根据你描述的症状，可能存在肠胃疾病的情况。此外，饮食习惯也会影响到这种情况发生。从中医角度来看，这主要与脾胃功能失调有关。因为脾胃相表里，所以需要通过健脾胃的方法进行治疗。同时，平常饮食也需注意清淡易消化。

(Hello! Based on the symptoms you described, there might be a potential issue with your gastrointestinal system. Moreover, dietary habits can also affect the occurrence of such symptoms. From a traditional Chinese medicine perspective, this is primarily related to a disorder of spleen and stomach functions, since they are closely linked. Therefore, treatment would involve methods to strengthen the spleen and stomach. Additionally, you should consider a light and easily digestible diet in your daily eating habits.)

---

**HuatuoGPT (w/ distilled data):**

您好，感谢您向我咨询。根据您所提供的症状描述，可能存在消化道炎症或者肠胃功能紊乱等情况。

首先，为了缓解腹泻症状，请注意以下几点：

1.多饮清水，避免脱水；

2.少量多餐，保持营养均衡，避免过饱或饿肚子；

3.避免摄入刺激性食物，例如辣椒、生姜等；

4.适当休息，避免剧烈运动和疲劳。

其次，关于您目前所说的牛奶可能引起腹泻的问题，需要进一步了解您的个人情况才能做出判断。通常来讲，牛奶中含有丰富的乳糖和蛋白质，而某些人群由于缺乏相应的酶类会产生乳糖不耐受或者蛋白质过敏等现象，从而引发腹泻。但也有很多人可以正常地饮用牛奶并且没有这种反应。因此，建议您再观察一段时间，看看症状是否有改善，如果仍然持续，建议到医院就诊，排除其他疾病后，如果需要，医生会给您更具体的治疗方案。

(Hello, thank you for reaching out. Based on the symptoms you've described, there could be inflammation in the digestive tract or some form of gastrointestinal dysfunction.

Firstly, to alleviate your diarrhea symptoms, please pay attention to the following points:

1. Drink plenty of clear fluids to prevent dehydration.

2. Eat small, frequent meals to maintain a balanced diet and avoid overeating or hunger.

3. Avoid foods that can irritate the digestive system, such as chili peppers and raw ginger.

4. Take adequate rest and avoid strenuous exercise and fatigue.

Secondly, regarding the possibility you mentioned about milk causing diarrhea, further information about your personal circumstances would be needed to make a judgement. Typically, milk contains rich lactose and proteins. Certain individuals lack the enzymes to digest these substances, leading to lactose intolerance or protein allergies that can cause diarrhea. However, many people can consume milk normally without such reactions. Therefore, I suggest you continue to monitor your symptoms for some time to see if there is any improvement. If symptoms persist, I recommend you seek medical attention. Once other diseases have been ruled out, if necessary, your doctor can provide you with a more specific treatment plan.)

---

**HuatuoGPT (w/o RLMF):**

您好，根据您提供的症状描述，很可能是肠胃炎或者消化系统感染引起的腹泻。而泡沫黄色液体通常表示胆汁排泄过多，这也与您所说的饮食习惯（比如喝牛奶）相关联。

为了更好地帮助您做出正确的诊断并获得恰当的治疗方案，请告诉我以下信息：

1. 您的年龄和性别？

2. 这种情况持续了多长时间？

3. 除了拉肚子外，还有没有其他不适症状，例如呕吐、发热等？

4. 您目前的饮食习惯和生活方式是怎样的？

5. 您是否曾经接受过类似的治疗？

(Hello, according to the description of the symptoms you provided, it is likely that the diarrhea has been caused The frothy yellow fluid usually indicates excessive bile excretion, which is also associated with the dietary habits you mentioned (such as drinking milk).

To better help you make the correct diagnosis and get the proper treatment plan, please tell me the following information:

1. your age and gender?

2. how long has this condition lasted?

3. Besides diarrhea, are there any other symptoms, such as vomiting or fever?

4. What are your current eating habits and lifestyle?

5. Have you ever received similar treatment?)

---

**HuatuoGPT:**

您好，根据您提供的症状描述，可能存在肠胃炎或消化不良等情况，而且泡沫黄色液体也表明可能存在某种程度的腹泻。关于饮食方面，有些人可能会对牛奶过敏或者容易引起腹胀、腹泻等反应，但具体还需要进行个性化评估才能得出结论。

为了更精确地判断您的病情，请问您以下几个问题：

1. 您现在多久拉一次？

2. 您每天摄入的水量和食物成分是否发生变化？

3. 除了拉肚子外，您是否还有其他不适感觉，比如恶心、呕吐、腹痛等？

4. 您平时是否吃较油腻或辛辣刺激性食品？

5. 您目前是否服用药物？

在获取到这些信息后，我将为您做出更加精准的诊断并给予相应的处置建议。

(Hello, based on the description of the symptoms you provided, there may be conditions such as gastroenteritis or indigestion, and the foamy yellow fluid also indicates that there may be some degree of diarrhea. Regarding diet, some people may be allergic to milk or prone to reactions such as bloating and diarrhea, but an individualized assessment is needed to reach a conclusion.

In order to determine your condition more precisely, please ask you the following questions:

1. how often are you pooping now?

2. Are there any changes in your daily water intake and food composition?

3. Besides diarrhea, do you have any other discomfort, such as nausea, vomiting, abdominal pain, etc.?

4. Do you usually eat fatty or spicy food?

5. Are you currently taking any medications?

After obtaining this information, I will make a more accurate diagnosis and give you appropriate treatment advice.)

Table 9: Sample Responses for Ablation Study.

| Comparison | Win | Lose |
|---|---|---|
| Single-turn | | |
| **HuatuoGPT-7B** vs ChatGPT | 52% | 48% |
| Multi-turn | | |
| **HuatuoGPT-7B** vs ChatGPT | 58% | 42% |

Table 10: Manual Evaluation of the Model Performance of 7B Scale.