# OpenReview forum: "HuatuoGPT, Towards Taming Language Model to Be a Doctor"
_EMNLP/2023/Conference — EMNLP 2023 Findings_

### Official Review · Reviewer_WjkR · 2023-07-31

**Soundness:** 4

**Excitement:**

3: Ambivalent: It has merits (e.g., it reports state-of-the-art results, the idea is nice), but there are key weaknesses (e.g., it describes incremental work), and it can significantly benefit from another round of revision. However, I won't object to accepting it if my co-reviewers champion it.

**Missing References:**

The paper is about taming Language Models To Be a *Doctor*. But the eval is only on doctors during *online* consultations. You should also consider a recent [competition](https://github.com/abachaa/MEDIQA-Chat-2023) that is *real-world* face-to-face consultation.

**Paper Topic And Main Contributions:**

This paper is about HuatuoGPT, a large language model for medical consultation in Chinese. The paper claims that HuatuoGPT can leverage both distilled data from ChatGPT and real-world data from doctors to generate patient-friendly and doctor-like responses. The paper also proposes a reinforcement learning method to further improve the model’s performance based on mixed feedback from ChatGPT and doctors. The paper evaluates HuatuoGPT on several Chinese medical benchmarks to show its effectiveness in roleplaying internet doctors and medical QA dataset, Experiment results show that HuatuoGPT outperforms existing open-source medical models and even ChatGPT in most cases.

**Questions For The Authors:**

E What is the [computation budget](https://aclanthology.org/D19-1224/) to train HuatuoGPT?

F Why HuatuoGPT performs similarly on Multi-turn Conversations compared to HuatuoGPT (w/ real) 0.66 vs 0.64, but so differently in Single-turn Conversations (0.575 vs 0.385) ? Could the reason be that your human data is multiround format only (no single round)?

G Under multi-turn evaluation, you evaluate on the results from doctor model given patient input. But could patient-chatgpt hallucinate and produce irrelevant answer/question that may lead to a bad question/answer generated by the doctor model?

**Reasons To Accept:**

A The author supervised fine-tuned (SFT) a model which combined the strengths of both distilled data from ChatGPT and real-world data from doctors to train a medical large language model.

B Other than previous hasty doctorbot models which only include SFT, the author applied a modified RLHF from the InstructGPT paper after SFT. They used mixed feedback from ChatGPT and doctors to reward the model for generating patient-friendly and doctor-like responses.

C The author presented a comprehensive evaluation protocol that covered single-turn and multi-turn medical consultations, with both automatic (GPT-4 as evaluator) and human evaluation.

D Ablation study shows that HuatuoGPT (w/real doctor data) performs better in multi-turn conversations while HuatuoGPT(w/ gpt distilled data) performs better in single-turn qa.

**Reasons To Reject:**

A Missing definition on multi-turn conversations *task* around line327. What is the input and expected output of this task. From figure 1, it seems you only have conversation history and patient-chatgpt’s utterance as input for doctor-model, but from the example in page 21, it seems like doctor-model also have access to clinical note (or patient case description) as doctor model’s input. Please specify the exact input and output for patient-chatgpt and doctor-model to help increase reproducibility.

B You claim that Doctors’ Responses are not patient-friendly (short and etc) as your motivation in line145, and thus to train using gpt distilled data. While this may intuitively be true, but does training with gpt distilled data help improve patient-friendliness, any experiment result to show this?

C Is gpt distilled data really needed? You hypothesis that GPT-4 exhibits poor performance in medicine compared to domain experts in Line048, which inspired you to use gpt distilled data. Any citation to support your hypothesis? From [here](https://arxiv.org/pdf/2303.13375.pdf), It seems like GPT-4 scored 92% on the USMLE Step 1 exam, surpassing the passing score of 75% by a large margin, which contradicts with your claim. Also your ablation study in figure 5, HuatuoGPT without gpt distilled data (w/ real only) did not differ much (only 0.02) from your HuatuoGPT on Multi-turn Conversations win-rate metric, especially when compared to huge gap between HuatuoGPT (w/ dist only) and HuatuoGPT (0.23).

D Even say if the 0.02 improvement is indeed statistically significant in previous point C, would it be cost-effective, as the training data (w/ real) is almost the same as (w/ dist)? train twice the amount of time to just improve 0.02?

**Reproducibility:**

4: Could mostly reproduce the results, but there may be some variation because of sample variance or minor variations in their interpretation of the protocol or method.

**Reviewer Confidence:**

4: Quite sure. I tried to check the important points carefully. It's unlikely, though conceivable, that I missed something that should affect my ratings.

**Typos Grammar Style And Presentation Improvements:**

Med-PaLM2 instead of Med-PaML2 in Table 6.

Tense error in “we *will* randomly switch the position of two responses.” in line354

---

> ### Author Rebuttal · Authors · 2023-08-29
>
> We would like to express our sincere gratitude for your time and effort in reviewing our manuscript. Your insights are invaluable, and we appreciate the opportunity to clarify any ambiguities. Please find our responses to your comments below.
>
> > **Q1: Missing definition on multi-turn conversations task around line327. What is the input and expected output of this task. From figure 1, it seems you only have conversation history and patient-chatgpt’s utterance as input for doctor-model, but from the example in page 21, it seems like doctor-model also have access to clinical note (or patient case description) as doctor model’s input. Please specify the exact input and output for patient-chatgpt and doctor-model to help increase reproducibility.**
>
> **Reply**: Thank you for pointing out the lack of clarity in our description of the multi-turn conversations task. We understand the importance of clearly specifying each step to ensure reproducibility. Here's a more detailed breakdown:
>
> **Input for Patient-ChatGPT**:
> Patient-ChatGPT is provided with the "patient case description," which serves as a background to help it simulate the role of a patient effectively, as mentioned in L327. After processing the patient case description, Patient-ChatGPT produces a simulated patient utterance.
>
> **Input for Doctor-Model**:
> The Doctor-Model receives the conversation history and the simulated patient utterance from Patient-ChatGPT as its input. Contrary to what might have been implied on page 21, the Doctor-Model does not have access to the clinical note or the patient case description.
>
> The example on page 21 was intended to showcase the background against which ChatGPT simulated the patient role, but it wasn't the direct input for the Doctor-Model. We'll work on clarifying this distinction in our revised version to prevent any confusion.
>
>
> > **Q2: You claim that Doctors’ Responses are not patient-friendly (short and etc) as your motivation in line145, and thus to train using gpt distilled data. While this may intuitively be true, but does training with gpt distilled data help improve patient-friendliness, any experiment result to show this?**
>
> **Reply**: We completely agree with the reviewer that it is essential to provide evidence for our claims. Here's a more detailed breakdown of our evidence:
>
> **Guidelines for Review**: As mentioned in L341, when we conducted evaluations using GPT4, one of our key metrics was the assessment of patient-friendliness. This is to ensure that the responses generated by the model met the patient's comfort and understanding needs.
>
> **Empirical Evidence**: We have presented a comprehensive case study in Table 9 in the appendix. A clear observation from this table is that the responses created using ChatGPT distillation data are more expansive and explanatory, making them more patient-friendly compared to the responses from doctors. To illustrate, consider the following examples in Table 9:
> | Content |
> |-------|
> | **User**: 我最近拉肚子，有泡沫黄水，怀疑是喝牛奶喝的，是不是喝牛奶的原因。谢谢。 (I’ve been experiencing diarrhea recently, with my stool appearing as frothy yellow liquid. I suspect it might be due to the milk I’ve been drinking. Could this be the reason? Thank you.)  |
> | **HuatuoGPT (w/ real data)**: 您好！根据你描述的症状，可能存在肠胃疾病的情况。此外，饮食习惯也会影响到这种情况发生。从中医角度来看，这主要与脾 胃功能失调有关。因为脾胃相表里，所以需要通过健脾胃的方法进行治疗。同时，平常饮食也需注意清淡易消化。 (Hello! Based on the symptoms you described, there might be a potential issue with your gastrointestinal system. Moreover, dietary habits can also affect the occurrence of such symptoms. From a traditional Chinese medicine perspective, this is primarily related to a disorder of spleen and stomach functions, since they are closely linked. Therefore, treatment would involve methods to strengthen the spleen and stomach. Additionally, you should consider a light and easily digestible diet in your daily eating habits.)  |
> | **HuatuoGPT (w/ distilled data)**: 您好，感谢您向我咨询。根据您所提供的症状描述，可能存在消化道炎症或者肠胃功能紊乱等情况。 首先，为了缓解腹泻症状，请注意以下几点：
> 1. 多饮清水，避免脱水；
> 2. 少量多餐，保持营养均衡，避免过饱或饿肚子；
> 3. 避免摄入刺激性食物，例如辣椒、生姜等；
> 4. 适当休息，避免剧烈运动和疲劳。
>
> 其次，关于您目前所说的牛奶可能引起腹泻的问题，需要进一步了解您的个人情况才能做出判断。通常来讲，牛奶中含有丰富的 乳糖和蛋白质，而某些人群由于缺乏相应的酶类会产生乳糖不耐受或者蛋白质过敏等现象，从而引发腹泻。但也有很多人可以正 常地饮用牛奶并且没有这种反应。因此，建议您再观察一段时间，看看症状是否有改善，如果仍然持续，建议到医院就诊，排除 其他疾病后，如果需要，医生会给您更具体的治疗方案。
>  (Hello, thank you for reaching out. Based on the symptoms you’ve described, there could be inflammation in the digestive tract or some form of gastrointestinal dysfunction.
> Firstly, to alleviate your diarrhea symptoms, please pay attention to the following points:
> 1. Drink plenty of clear fluids to prevent dehydration.
> 2. Eat small, frequent meals to maintain a balanced diet and avoid overeating or hunger.
> 3. Avoid foods that can irritate the digestive system, such as chili peppers and raw ginger.
> 4. Take adequate rest and avoid strenuous exercise and fatigue.
> Secondly, regarding the possibility you mentioned about milk causing diarrhea, further information about your personal circumstances would be needed to make a judgement. Typically, milk contains rich lactose and proteins. Certain individuals lack the enzymes to digest these substances, leading to lactose intolerance or protein allergies that can cause diarrhea. However, many people can consume milk normally without such reactions. Therefore, I suggest you continue to monitor your symptoms for some time to see if there is any improvement. If symptoms persist, I recommend you seek medical attention. Once other diseases have been ruled out, if necessary, your doctor can provide you with a more specific treatment plan.)
>
> > **Q3: Is gpt distilled data really needed? You hypothesis that GPT-4 exhibits poor performance in medicine compared to domain experts in Line048, which inspired you to use gpt distilled data. Any citation to support your hypothesis? From here, It seems like GPT-4 scored 92% on the USMLE Step 1 exam, surpassing the passing score of 75% by a large margin, which contradicts with your claim. Also your ablation study in figure 5, HuatuoGPT without gpt distilled data (w/ real only) did not differ much (only 0.02) from your HuatuoGPT on Multi-turn Conversations win-rate metric, especially when compared to huge gap between HuatuoGPT (w/ dist only) and HuatuoGPT (0.23).**
>
> **Reply**:  We firmly believe that ChatGPT distilled data is necessary as it enables models to provide detailed, well-organized, and rich-format content. As can be seen in Table 9 or provided examples in Question B, HuatuoGPT with only real data provides shorter, less organized answers compared to HuatuoGPT (w/ distilled data). Despite the latter providing more general answers, it exhibits doctor-like behavior by offering professional suggestions and follow-up questions.
>
> Although GPT-4 has performed well on the USMLE Step 1 exam, it does not perform as well as expert doctors in practice, as shared by doctors we have consulted. Additionally, there are very few people who consult GPT instead of an expert doctor, highlighting its limitations. Moreover, our HuatuoGPT model is trained in the Chinese language, and GPT-4 does not pass most examinations in Chinese, as indicated in [the study](https://arxiv.org/pdf/2308.08833.pdf).
>
>
> > **Q4:  Even say if the 0.02 improvement is indeed statistically significant in previous point C, would it be cost-effective, as the training data (w/ real) is almost the same as (w/ dist)? train twice the amount of time to just improve 0.02?**
>
> **Reply**:  We acknowledge the reviewer's concern about cost-effectiveness. During the instruction fine-tuning stage, the number of training examples or steps does not significantly affect the model's quality. For instance, [LIMA](https://arxiv.org/pdf/2305.11206.pdf) only uses 1000 examples for instruction fine-tuning. The behavior differences shown in the table could not be attributed to scaling up the instruction. Importantly, using distilled data (w/ dist) and real-world data (w/ real) exhibit distinct advantages in Single-turn Questions and Multi-turn Conversations, respectively. Combining both improves performance in both areas, indicating their complementarity.
>
> We will clarify this point in the revised version.
>
>
>
> > **Q5: What is the computation budget to train HuatuoGPT?**
>
> **Reply**: Training HuatuoGPT requires approximately 200 A100-GPU hours, with 120 A100-GPU hours consumed during the SFT stage and about 80 A100-GPU hours consumed during the RLMF stage.
>
>
> > **Q6: Why HuatuoGPT performs similarly on Multi-turn Conversations compared to HuatuoGPT (w/ real) 0.66 vs 0.64, but so differently in Single-turn Conversations (0.575 vs 0.385) ? Could the reason be that your human data is multiround format only (no single round)?**
>
> **Reply**: Our real-world data includes both single and multi-round formats. However, as discussed in the paper, real-world data produces overly concise responses and lacks the necessary level of detail due to the time constraints faced by doctors during the diagnostic process. On the other hand, distilled data and ChatGPT generate detailed and well-presented responses. Therefore, HuatuoGPT (w/ real) learns the diagnostic ability of doctors to some extent, but its responses are neither detailed nor well-orgainzed, leading to its subpar performance in single-turn conversations. This underscores the need to use both real-world and distilled data.
>
>
> > **Q7: Under multi-turn evaluation, you evaluate on the results from doctor model given patient input. But could patient-chatgpt hallucinate and produce irrelevant answer/question that may lead to a bad question/answer generated by the doctor model?**
>
> **Reply**:  Indeed, ChatGPT, as a patient-chatgpt, may produce hallucinated or irrelevant answers/questions. However, in our experiments, we found that ChatGPT effectively generated questions with our well-designed prompt based on the provided patient information. It also responded accurately to the doctor's questions, with very few instances of asking irrelevant questions that could mislead the doctor model. We will include examples and an intriguing research direction for the future, and we leave this issue for further studies.
>
> Thank you again for your insightful comments and questions. We believe that addressing these points will significantly improve our work, and we are grateful for the opportunity to clarify these aspects.

---

### Official Review · Reviewer_Dcp8 · 2023-08-04

**Soundness:** 3

**Ethical Concerns:**

Yes

**Excitement:**

3: Ambivalent: It has merits (e.g., it reports state-of-the-art results, the idea is nice), but there are key weaknesses (e.g., it describes incremental work), and it can significantly benefit from another round of revision. However, I won't object to accepting it if my co-reviewers champion it.

**Justification For Ethical Concerns:**

This paper proposes a doctor-like language model to give a diagnosis. However, the data used for fine-tuning is partially extracted from ChatGPT, without any manual filtering or verification. While ChatGPT is a strong model, its efficacy as a medical diagnostic tool remains untested. This raises the question of how we can validate the accuracy of the derivative model, Huatuo. It's crucial to exercise extreme caution in this particular field.

While I concur with the ethical statements put forth by the authors, I harbor concerns about potential misuse and serious risks associated with this model. Perhaps obtaining an Institutional Review Board (IRB) license could mitigate these risks to a certain extent.

**Paper Topic And Main Contributions:**

This paper proposes HuatuoGPT, a highly effective medical model. It conducts supervised fine-tuning using both ChatGPT-distilled data and real-world medical data. Additionally, a mixed-feedback reward modeling approach is designed. The superiority of this model is demonstrated through automated evaluation with GPT4 and evaluation by expert doctors, outperforming numerous open-source models and ChatGPT. At the same time, great results are also achieved across various benchmarks.

**Questions For The Authors:**

Although the win/loss ratios between the models are shown in Figure 4, I'm more curious about the diagnostic accuracy of each model.

**Reasons To Accept:**

- With supervised fine-tuning and reinforcement learning, the model's performance seems good, validating that this paradigm can be implemented well in the medical field.
- The medical SFT data and pair-wise comparison data are valuable.

**Reasons To Reject:**

- Test set is limited, automated evaluation using GPT-4 may be able to use a larger test set
- Even though the data from different sources were processed by means of role-play and LLM-polished, their overall distributions are still very different. So how to ensure that a user-friendly response format and professional guidance are integrated after training, instead of adopting appropriate response formats for different data distributions.
- There is too little description of the data, such as the exact amount of data, length, etc.
- This paper appears to be incremental, with a self-instruct method used for data construction, and RL with Mixed Feedback actually just includes real responses from doctors in the scoring, and the evaluation method is just based on GPT-4 comparisons.
- The presentation of Figure 5 is not very clear, does w/real mean SFT with real data only? Also, maybe add an experiment comparing HuatuoGPT and w/o RLMF, which would better show how much of a boost RLMF is.
- Suggesting to include a comparison between RLMF and RL (without reference when comparing) in the ablation experiments, I'm curious how much difference there is between the two?

**Reproducibility:**

4: Could mostly reproduce the results, but there may be some variation because of sample variance or minor variations in their interpretation of the protocol or method.

**Reviewer Confidence:**

4: Quite sure. I tried to check the important points carefully. It's unlikely, though conceivable, that I missed something that should affect my ratings.

---

> ### Author Rebuttal · Authors · 2023-08-29
>
> > **Q1.Test set is limited, automated evaluation using GPT-4 may be able to use a larger test set.**
>
> **Reply**: We understand the concern about the limited test set. Our GPT-4 quota constraints restricted us to 100 questions for testing before the submission. With an increased quota after that, we expanded our tests to 500 questions from KUAKE-QIC instead of the original 100 questions. The extended results show that HuatuoGPT still wins ChatGLM-6b, Ziya-LLaMA-13B and ChatGPT; this confirms the validity of our model as the dataset size grows. We appreciate this suggestion and will incorporate these extended results in the revised version.
>
> | HuatuoGPT Win rate                            | 100 Questions (win/ tie/ fail) | 500 Questions (win/ tie/ fail) |
> |-------------------------------------|--------------------------------|--------------------------------|
> | HuatuoGPT vs ChatGLM-6b             | **43** / 18 / 39                   | **251** / 59 / 190                 |
> | HuatuoGPT vs Ziya-LLaMA-13B         | **50** / 13 / 37                   | **260** / 66 / 174                 |
> | HuatuoGPT vs ChatGPT                | **47** / 21 / 32                   | **234** / 55 / 211                 |
>
> > **Q2: Even though the data from different sources were processed by means of role-play and LLM-polished, their overall distributions are still very different. So how to ensure that a user-friendly response format and professional guidance are integrated after training, instead of adopting appropriate response formats for different data distributions.**
>
> **Reply**: We acknowledge the difference in distributions between the two groups data. The two means during SFT (i.e., role-play and LLM-polishing) could help it enjoy the both merits; but it is not guaranteed. To address this,
> we employed the RLMF to ensure the effective integration of a user-friendly response format and professional guidance after STF training. Technically, RLMF adopts the prompt of ChatGPT to emphasize a user-friendly response format while using doctors' references to reward consistency with professional doctors. Therefore, only the response that satisfies both feedback preferences can receive higher rewards.
>
> > **Q3. There is too little description of the data, such as the exact amount of data, length, etc.**
>
> **Reply**: We apologize for the oversight, but we incorporate the description in the appendix. For comprehensive data details, please refer to Table 3 in Appendix C.
>
> | Data Type       | Source                        | Number of Samples |
> |-----------------|-------------------------------|-------------------|
> | Instruction     | ChatGPT-generated                      | 61,400            |
> | Instruction     | Real-world | 69,768            |
> | Conversation    | ChatGPT-generated                       | 69,888            |
> | Conversation    | Real-world | 25,986  |
>
> It provides a breakdown of the roughly 230,000 data points used in the SFT process, indicating the exact amount from each source. In the Instruction data, 61400 samples are distilled from ChatGPT and 69768 samples from real-world doctor-patient QAs. In Conversation data, 69888 samples are distilled from ChatGPT and 25986 samples from real-world doctor-patient conversations. We opted to place this table in the appendix due to limited space. We will incorporate more data specifics, as you've suggested, in our revised revision.
>
> > **Q4. This paper appears to be incremental, with a self-instruct method used for data construction, and RL with Mixed Feedback actually just includes real responses from doctors in the scoring, and the evaluation method is just based on GPT-4 comparisons.**
>
> **Reply**: Thank you for the feedback. Our main contribution lies in introducing an approach to effectively enjoy both merits of self-instruction data with real-world data, in a common two-stage training paradigm, which makes the HuatuoGPT model inherit the advantages of ChatGPT and the doctor to generate user-friendly and professional responses.
>
> **Data level**:  During the SFT stage, we propose a role-playing-based self-instruction and LLM-Polished method to narrow the gap between the distilled data and real-world data, ensuring a smoother integration of diverse data.
>
> **Reward level**: In the RL stage, we propose a mixed feedback RL method that aims for responses to be both patient-friendly and doctor-like through two sources of feedback: the prompt of ChatGPT and doctor references. It enables our model to integrate the two groups' data better because only the generated response that satisfies both feedback preferences can receive higher rewards.
>
> The experimental results show that our model significantly outperforms the existing models in both automatic and manual evaluations, demonstrating the effectiveness of our method.
>
> For evaluation, we also incorporate human evaluations and medical QA other than GPT-4 evolutions. This makes our assessment one of the most comprehensive evaluation protocols compared to concurrent works [e.g., Ziya, DoctorGLM, Bencao]. We acknowledge the challenges in evaluating medical language models.
>
> > **Q5. The presentation of Figure 5 is not very clear, does w/real mean SFT with real data only? Also, maybe add an experiment comparing HuatuoGPT and w/o RLMF, which would better show how much of a boost RLMF is.**
>
> **Reply**: We apologize for the confusion. "w/real" in Figure 5 refers to SFT conducted using only real data, which is a HuatuoGPT variant without distilled data and RLMF. Regarding your suggestion for a comparison between HuatuoGPT and w/o RLMF, we have conducted an additional experiment comparing the responses of GPT-4, HuatuoGPT, and w/o RLMF. The results in the following table show HuatuoGPT outperforms w/o RLMF.
>
> | HuatuoGPT Win rate                                        | Single turn              | Multi turn               |
> |-------------------------------------------------|--------------------------|--------------------------|
> | HuatuoGPT vs HuatuoGPT w/o RLMF| **51** / 26/ 23              | **49**/ 12/ 39               |
>
> **Q6: Suggesting to include a comparison between RLMF and RL (without reference when comparing) in the ablation experiments, I'm curious how much difference there is between the two?**
>
> **Reply**: Thank you for your constructive suggestion. We have added an experiment comparing HuatuoGPT between the RLMF and the RL without doctor's references. The results show using RLMF is better than using RL without doctor's reference, demonstrating the effectiveness of our method:
>
> | RLMF Win rate                          | Single turn  (win/ tie/ faile)                | Multi turn  (win/ tie/ faile)
> |-----------------------------------|-----------------------------|----------------------------|
> |  RLMF vs RL w/o doctor reference | **46** / 14/ 40                 | **60**/ 7/ 33                  |
>
> **Q7: Although the win/loss ratios between the models are shown in Figure 4, I'm more curious about the diagnostic accuracy of each model.**
>
> **Reply**: To address your curiosity about diagnostic accuracy, we had doctors compare the diagnostic accuracy of the model's output on the Multi-turn Conversations of the experiments. In our experiments, we score the accuracy of the final advice given by the model and follow the following guidelines:
> 1. Since it is a chatgpt simulation of a patient, we do not include in our calculations when the dialogue deviates from the original diagnosis and treatment context. We also do not include data in the calculation if the doctor did not give a clear conclusion in the real diagnosis.
> 2. For the patient's diagnosis, we considered it accurate if the model mentioned the same diagnosis as the real doctor.
> 3. For the treatment recommendation of a known condition, we consider it accurate if the model mentions the same treatment recommendation as the real doctor.
>
> The diagnostic accuracy rates for different models are as follows:
>
> | Models                          | diagnostic accuracy                |
> |-----------------------------------|-----------------------------|
> |  ChatGLM   |    0.23              |
> |  Ziya-LLaMA-13B   |   0.26           |
> |  ChatGPT   |   0.40               |
> |  HuatuoGPT   | 0.47             |
>
> As can be seen, HuatuoGPT has significantly higher diagnostic accuracy than other models, though there is still room for improvement. With ChatGLM and Ziya, they can usually give very informative answers, but they are always too broad to get to the point. For ChatGPT, it has strong instruction following ability, but it is usually hard to give a clear diagnosis. We think these are the strengths of HuatuoGPT.
> Furthermore, it should be noticed that doctors will be very concerned about the accuracy of the diagnosis during the evaluation, which is the guideline we mentioned.
>
> ##  Ethical Concerns:
> > **Q8: Concerns about the use of ChatGPT data**
>
> **Reply**: Thank you for raising concerns about the use of ChatGPT data.
>
> We recognize the potential limitations of using ChatGPT as a data source: We primarily utilized ChatGPT to capture the format and tone suitable for structured and comprehensive responses, not as a source of medical accuracy.
>
> To ensure medical correctness, we complemented this with training data from real doctors in the SFT with the Hybrid Data stage. In addition, real doctor responses also served as benchmarks during feedback collection from ChatGPT in the RL with Mixed Feedback stage.
>
> We'd like to emphasize that our research is exploratory and for academic insights. We have no intention of deploying the model for medical practice due to its current diagnostic limitations.
>
>
> > **Q9: Concerns about potential misuse and associated risks**
>
> **Reply**: We share your concerns about potential misuse and risks.
>
> We fully recognize the importance of ethical considerations in deploying and distributing models like HuatuoGPT.
>
> To safeguard against misuse, we are contemplating restricted access, allowing only the scientific and professional communities to use our model.
>
> Additionally, our research has been reviewed and approved by our institute's ethics board, ensuring it aligns with established ethical guidelines.
>
> We deeply appreciate your concerns and are committed to taking all necessary precautions. Your feedback is invaluable, and we sincerely thank you for helping us enhance the integrity and safety of our work.

---

### Official Review · Reviewer_rXzS · 2023-08-05

**Soundness:** 2

**Excitement:**

3: Ambivalent: It has merits (e.g., it reports state-of-the-art results, the idea is nice), but there are key weaknesses (e.g., it describes incremental work), and it can significantly benefit from another round of revision. However, I won't object to accepting it if my co-reviewers champion it.

**Paper Topic And Main Contributions:**

This paper presents a large language model for Chinese medical consultation, HuatuoGPT, which leverages both distilled data from ChatGPT and real-world data from doctors in the SFT stage. To further synergize the strengths of both data sources,  they also adopt RLMF strategy for RL stage. Experiments are conducted on single-turn/multi-turn medical dialogue datasets and medical QA benchmarks, and achieve SOTA compared with open-source LLMs.

**Questions For The Authors:**

1. In line 335, "When evaluated by doctors, we randomly sample 50 patient cases from 100 test cases", but in Figure 4(b), the results still contain 100 cases.
2. For medical QA, how are the results of fine-tuning the HuatuoGPT by using training data?

**Reasons To Accept:**

1. This paper develops a Chinese medical LLM, using data from both ChatGPT and doctor and adopting RLMF training stategy.
2. Compared with other open-source LLMs, HuatuoGPT performs better on medical dialogue and QA.

**Reasons To Reject:**

1. The contribution of this paper mostly lies in engineering.
2. Since the experiements of single-turn/multi-turn dialogue generation are conducted on a filtered small test sets, and the experimental results of medical QA benchmarks are mostly worse than T5 (finetuned) except for cMedQA2, so the value of the experimental results doesn't seem to be significant.

**Reproducibility:**

4: Could mostly reproduce the results, but there may be some variation because of sample variance or minor variations in their interpretation of the protocol or method.

**Reviewer Confidence:**

4: Quite sure. I tried to check the important points carefully. It's unlikely, though conceivable, that I missed something that should affect my ratings.

---

> ### Author Rebuttal · Authors · 2023-08-29
>
> We sincerely thank the reviewer for their insightful comments and observations.
>
> > **Q1.The contribution of this paper mostly lies in engineering**
>
> **Reply**: We acknowledge the reviewer's point about the engineering aspect of our work. Our research philosophy is to first investigate a straightforward and simple approach. Such an approach, especially in the era of large language models, often leans toward partially-engineering solutions. We believe that engineering-assisted research, as a first step, is beneficial for contextualizing the potential of large language models in medical scenarios. Specifically, this may provide valuable insights for the audience from the EMNLP (**Empirical** Methods in Natural Language Processing) community .
>
> > **Q2. Since the experiments of single-turn/multi-turn dialogue generation are conducted on filtered small test sets, and the experimental results of medical QA benchmarks are mostly worse than T5 (finetuned) except for cMedQA2, the value of the experimental results doesn't seem to be significant.**
>
> **Reply**: We appreciate the feedback on our evaluation approach and recognize its limitations due to GPT-4 access quota constraints and the cost of doctor evaluations. We believe that the evaluation is reliable, as seen from the following aspects:
>
> - **Blind Evaluation**: Our evaluation approach, involving medical professionals, was blind. This means that the evaluating doctors weren't aware of the answer's source, ensuring unbiased feedback.
>
> - **Consistency between GPT-4 and doctors**: Despite the smaller test set, the doctors' evaluations aligned closely with our GPT-4 evaluations. This consistency suggests that our evaluations, although limited in scale, can be considered reliable.
>
>
> **A slightly larger test set**
> We understand the concern about the limited test set. Our GPT-4 quota constraints restricted us to 100 questions for testing before the submission. With an increased quota after that, we expanded our tests to 500 questions from KUAKE-QIC instead of the original 100 questions. The extended results show that HuatuoGPT still wins all baselines,  confirming the validity of our model as the dataset size grows. We appreciate this suggestion and will incorporate these extended results in the revised version.
>
> | HuatuoGPT Win rate                            | 100 Questions (win/ tie/ fail) | 500 Questions (win/ tie/ fail) |
> |-------------------------------------|--------------------------------|--------------------------------|
> | HuatuoGPT vs ChatGLM-6b             | **43** / 18 / 39                   | **251** / 59 / 190                 |
> | HuatuoGPT vs Ziya-LLaMA-13B         | **50** / 13 / 37                   | **260** / 66 / 174                 |
> | HuatuoGPT vs ChatGPT                | **47** / 21 / 32                   | **234** / 55 / 211                 |
>
> **Performance of QA benchmark**: While HuatuoGPT might lag behind the fine-tuned T5 on certain benchmarks, this comparison isn't strictly apples-to-apples. T5 is trained using a supervised dataset, a.k.a **full-shot** setting. While HuatuoGPT is evaluated in a zero-shot setting. Note that HuatuoGPT still outperforms the fine-tuned T5 in some metrics in all datasets (e.g., BLEU-1). Especially, in the zero-shot setting, HuatuoGPT outshines other chat models, demonstrating its versatility and commendable ability to perform out-of-distribution tasks. This distinction underscores the unique strengths of HuatuoGPT.
>
>
> > **Q3: In line 335, "When evaluated by doctors, we randomly sample 50 patient cases from 100 test cases", but in Figure 4(b), the results still contain 100 cases.**
>
> **Reply**: Apologies for the oversight. The discrepancy arises from how we presented the data. While we sampled 50 patient cases for doctor evaluation, Figure 4(b) displays results in **percentages** based on those 50 cases, making it appear as if it's out of 100. For exact numbers, please refer to Table 5. We'll ensure clearer representation in the revised version.
>
> > **Q4: For medical QA, how are the results of fine-tuning the HuatuoGPT by using training data?**
>
> **Reply**:
> Due to time limitations, we sampled 50,000 QA (20% training data) from the webMedQA training set to fine-tune HuatuoGPT. The following table shows fine-tuning HuatuoGPT with the webMedQA training set leads to an improvement in all metrics, indicating the transferability and generalization of HuatuoGPT. In addition, the table shows that even with 20% training data, HuatuoGPT achieves comparable performance or surpasses the T5 model with 100% training data in most indicators, demonstrating our model's potential. We will give more detailed experiments and analysis in the revised version.
>
> | Model                  | BLEU-1 | BLEU-2 | BLEU-3 | BLEU-4 | GLEU | ROUGE-1 | ROUGE-2 | ROUGE-L |
> |------------------------|--------|--------|--------|--------|------|---------|---------|---------|
> | Fine-tuned T5              | 21.42  | 13.79  | 10.06   | 7.38  | 8.94 | 31.00  | 13.85   |  25.78   |
> | HuatuoGPT              | 24.85  | 13.42  | 7.72   | 4.51   | 7.50 | 28.30   | 7.72    | 14.50   |
> | Fine-tuned HuatuoGPT（20% training data）| 25.05  | 14.15  | 9.23   | 6.32   | 7.56 | 30.89   | 9.55    | 20.95   |

---

### Meta-Review · Area_Chair_LkFf · 2023-09-19

**Recommendation:** 3

**Metareview:**

The paper introduces HuatuoGPT, a large language model trained for Chinese medical consultation tasks. The model leverages distilled data from ChatGPT and real-world medical data, along with a mixed-feedback reinforcement learning strategy (RLMF). The paper shows that the model performs well in both single-turn and multi-turn medical dialogue as well as medical QA tasks, often outperforming existing open-source large language models (LLMs). The reviewers generally agree that the paper is technically sound and reports state-of-the-art results. However, they express concerns about the paper's incremental contributions and lack of significant experimental validation. All reviewers highlighted that the work appears to be an engineering-focused extension of existing techniques rather than a novel scientific contribution. There are concerns about the limited size and representativeness of the test sets used for validation, which raises questions about the generalizability of the results. Considering the paper's technical soundness and its ability to contribute to the field in a meaningful, if incremental, manner, the paper is thus recommended to be accepted to the "Findings" track. To further enhance the paper's value, the authors may wish to consider addressing concerns raised by the reviewers.

---

### Decision · Program_Chairs · 2023-10-07

**Decision:**

Accept-Findings

**Comment:**

The paper introduces HuatuoGPT, a large language model trained for Chinese medical consultation tasks. The model leverages distilled data from ChatGPT and real-world medical data, along with a mixed-feedback reinforcement learning strategy (RLMF). The paper shows that the model performs well in both single-turn and multi-turn medical dialogue as well as medical QA tasks, often outperforming existing open-source large language models (LLMs). The reviewers generally agree that the paper is technically sound and reports state-of-the-art results. However, they express concerns about the paper's incremental contributions and lack of significant experimental validation. All reviewers highlighted that the work appears to be an engineering-focused extension of existing techniques rather than a novel scientific contribution. There are concerns about the limited size and representativeness of the test sets used for validation, which raises questions about the generalizability of the results. Considering the paper's technical soundness and its ability to contribute to the field in a meaningful, if incremental, manner, the paper is thus recommended to be accepted to the "Findings" track. To further enhance the paper's value, the authors may wish to consider addressing concerns raised by the reviewers.